# Tractable Representation Learning with Probabilistic Circuits

**Steven Braun**                                    *steven.braun@cs.tu-darmstadt.de*
*Technische Universität Darmstadt, Germany*

**Sahil Sidheekh**                                      *sahil.sidheekh@utdallas.edu*
*University of Texas at Dallas, United States*

**Antonio Vergari**                                            *avergari@ed.ac.uk*
*University of Edinburgh, United Kingdom*

**Martin Mundt**                                           *mundtm@uni-bremen.de*
*Universität Bremen, Germany*

**Sriraam Natarajan**                                 *sriraam.natarajan@utdallas.edu*
*University of Texas at Dallas, United States*

**Kristian Kersting**                              *kristian.kersting@cs.tu-darmstadt.de*
*Technische Universität Darmstadt, Germany*

**Reviewed on OpenReview:** *https://openreview.net/forum?id=h8D75pVKja*

## Abstract

Probabilistic circuits (PCs) are powerful probabilistic models that enable exact and tractable inference, making them highly suitable for probabilistic reasoning and inference tasks. While dominant in neural networks, representation learning with PCs remains underexplored, with prior approaches relying on external neural embeddings or activation-based encodings. To address this gap, we introduce autoencoding probabilistic circuits (APCs), a novel framework leveraging the tractability of PCs to model probabilistic embeddings explicitly. APCs extend PCs by jointly modeling data and embeddings, obtaining embedding representations through tractable probabilistic inference. The PC encoder in APCs delivers a hybrid neural autoencoding framework where missing data with arbitrary missing patterns can be natively handled in a principled way while keeping training end-to-end differentiable. Our empirical evaluation demonstrates that APCs outperform existing PC-based autoencoding methods in reconstruction quality, generate embeddings competitive with, and exhibit superior robustness in handling missing data compared to neural autoencoders. These results highlight APCs as a powerful and flexible representation learning method that exploits the probabilistic inference capabilities of PCs, showing promising directions for robust inference, out-of-distribution detection, and knowledge distillation.

## 1 Introduction

The ability to learn compact and expressive data representations is a fundamental aspect of modern machine learning. Representation learning involves automatically discovering and encoding informative, low-dimensional features or embeddings from raw data. These effectively capture underlying factors of variation to facilitate downstream tasks such as classification, generation, or retrieval. Autoencoders have played a pivotal role in this domain by enabling the discovery of low-dimensional embeddings that capture latent data features (Hinton & Salakhutdinov, 2006; Rifai et al., 2011). Their impact extends across diverse applications, such as self-supervised learning paradigms like joint embedding predictive architectures (Assran et al., 2023), natural language processing, where token embeddings are crucial for large language models (Vaswani et al.,

2017; Devlin et al., 2019), and computer vision, where embeddings facilitate efficient image retrieval in large databases (Deng et al., 2009). These advancements predominantly rely on neural network-based approaches, highlighting the critical role of embeddings in contemporary representation learning. Neural networks have been extensively studied in the field of representation learning, leading to their probabilistic extension, variational autoencoders (VAEs; Kingma & Welling, 2014; Rezende et al., 2014), later improved and extended along several dimensions such as the introduction of vector quantization (VQ-VAE; van den Oord et al., 2017; Razavi et al., 2019), and hierarchical (Vahdat & Kautz, 2020; Liévin et al., 2019) and deterministic (Ghosh et al., 2020) formulations.

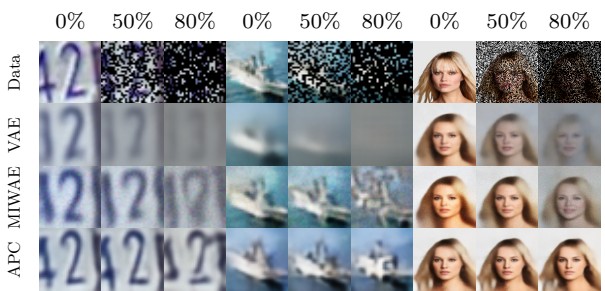

Figure 1: **APCs outperform VAE-based models in reconstruction quality under missing data.** For experiment settings, see Section 5.1.

In recent years, a special kind of neural network has emerged for tractable probabilistic modeling: probabilistic circuits (PCs; Darwiche, 2003b; Choi et al., 2020). The computational graphs of PCs are constrained in the way they are formed, thus trading off expressiveness for tractable inference (Vergari et al., 2019b). For example, given certain structural properties that are easy to enforce in practice, PCs can allow the tractable marginalization of *every* subset of the input features in just a single feedforward pass of the computational graph. These models provide a unified framework for tractable probabilistic models (Vergari et al., 2021) and address tractability challenges in deep generative models (Darwiche, 2003a; Poon & Domingos, 2011; Kisa et al., 2014). They have been successfully applied in diverse areas, including lossless compression (Liu et al., 2022; Severo et al., 2025), biomedical modeling (Dang et al., 2022b; Mathur et al., 2023; 2024), neuro-symbolic AI (Ahmed et al., 2022; Loconte et al., 2023; Karanam et al., 2025; Kurscheidt et al., 2025), multi-modal fusion (Sidheekh et al., 2025), graph representation and learning (Errica & Niepert, 2023; Zheng et al., 2018; Loconte et al., 2023), and constrained text generation (Zhang et al., 2023). Recent advancements have enhanced PCs in expressivity, learning, and scaling through sparsity-inducing regularization (Dang et al., 2022a), structural improvements through subtractive mixtures and squared circuits (Loconte et al., 2024; 2025b; Wang & Broeck, 2025), as well as hybrid approaches like HyperSPNs (Shih et al., 2021), neural conditioning in conditional sum-product networks (Shao et al., 2022) and probabilistic neural circuits (Martires, 2024), and latent variable distillation (Liu et al., 2023a;b). However, in contrast to neural networks, the use of PCs for representation learning has remained largely unexplored.

All the above works focus on learning or querying PCs as black-box probability models, i.e., functions that receive some input configuration and output a scalar probabilistic quantity of interest: a marginal or conditional probability or the value of an expectation. However, since PCs are neural networks, one could use them to extract representations and use these for downstream predictive tasks. To date, research into representation learning with PCs has been limited, with two notable exceptions. The first is the investigation of internal PC representations by Vergari et al. (2019a). The second is sum-product autoencoding (SPAE; Vergari et al., 2018), which defines an autoencoding scheme using these representations. However, these prior approaches face severe limitations: it is often unclear how to extract general representations decoupled from the specific circuit structure, and the autoencoding scheme cannot be trained end-to-end. This inherently prevents seamless integration with modern neural architectures.

We present *autoencoding probabilistic circuits* (APCs), a novel framework that introduces end-to-end learning of representations in an autoencoding fashion using PCs. APCs leverage the probabilistic capabilities of PCs by employing them as a tractable encoder, generating *explicit* probabilistic embeddings $\mathbf{Z}$. Critically, embeddings in APCs are obtained through tractable conditional state inference (i.e. sampling or maximum a posteriori) from the joint distribution $p_{\mathcal{C}}(\mathbf{X}, \mathbf{Z})$ over data $\mathbf{X}$ and embedding random variables $\mathbf{Z}$ explicitly modeled by the PC: $\mathbf{z} \sim p_{\mathcal{C}}(\mathbf{Z} \mid \mathbf{X}_o = \mathbf{x}_o)$, where $\mathbf{X}_o \subseteq \mathbf{X}$ can be partially observed, natively handling missing data. In stark contrast, this is hard to do with fully neural-based autoencoders, where people often resort to heuristics or computationally intensive imputation methods (Nazabal et al., 2020; Mattei & Frellsen, 2019; Simkus & Gutmann, 2023; 2024) to deal with incomplete inputs (see Section 4 and Fig. 1). APCs can

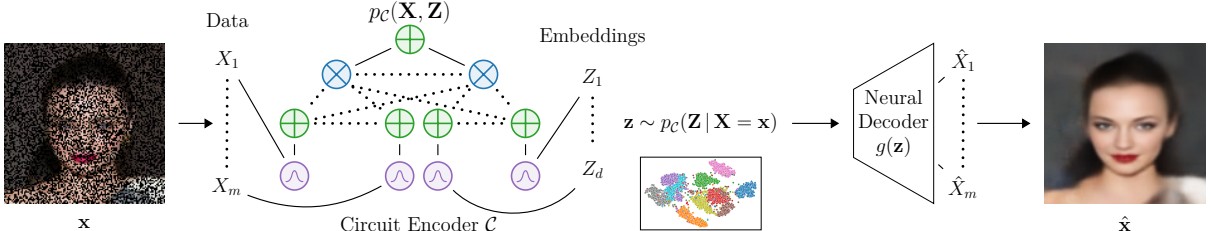

Figure 2: **Autoencoding probabilistic circuits (APCs) architecture**. An input data point, possibly with (arbitrary) missing values, is probabilistically encoded into a compact embedding space $\mathbf{Z}$ using a PC $p_{\mathcal{C}}(\mathbf{X}, \mathbf{Z})$. The embedding $\mathbf{z}$ is sampled from the data-conditional distribution $p_{\mathcal{C}}(\mathbf{Z} \mid \mathbf{X})$. A neural decoder then reconstructs the complete data point $\hat{\mathbf{X}}$ from the embedding. The tractable nature of the PC encoder allows the intrinsic handling of missing data without the need for imputation.

further bridge circuit and neural architectures by allowing for an arbitrary neural network decoder, extending the frameworks capacity. ***APCs thus offer an elegant and principled solution to the problem of learning representations in the presence of missing data***. Fig. 2 illustrates the APC pipeline.

In summary, our key contributions are: (a) autoencoding probabilistic circuits (APCs), a novel framework for tractable representation learning with explicit embeddings modeled with PCs; (b) integrating a hybrid architecture that improves the learning of the encoder circuit, combining the strengths of tractable probabilistic and neural methods; (c) enabling end-to-end training between the encoder circuit and neural decoder by an improved differentiable sampling procedure for PCs, leveraging novel advances in gradient estimation; and (d) extensively demonstrating APCs' ability to reconstruct inputs, produce useful embeddings for downstream tasks, and knowledge distillation without access to original training data, all in the presence of missing inputs, a scenario that neural autoencoders cannot natively handle.

## 2 Background

To properly introduce APCs later in Section 3, we need to briefly provide the necessary background about the mechanism of autoencoding and the formal definitions of PCs.

**Autoencoding.** The autoencoding mechanism, fundamental to representation learning and central to APCs, is a form of self-supervised learning (Hinton & Salakhutdinov, 2006). An autoencoder consists of two primary components: an *encoder* and a *decoder*. The encoder, $f : \mathbf{X} \mapsto \mathbf{Z}$, maps an input data point $\mathbf{x} \in \mathbf{X}$ to a typically lower-dimensional ($|\mathbf{Z}| \ll |\mathbf{X}|$) embedding representation, also called *embedding*, $\mathbf{z} \in \mathbf{Z}$. The decoder, $g : \mathbf{Z} \mapsto \mathbf{X}$, then attempts to reconstruct the original input from this embedding, producing $\hat{\mathbf{x}} = g(\mathbf{z}) = g(f(\mathbf{x}))$. The model is trained by minimizing a possibly regularized reconstruction loss, $\mathcal{L}(\mathbf{x}, \hat{\mathbf{x}})$, which measures the dissimilarity between the original input and its reconstruction. This process encourages the embedding space $\mathbf{Z}$ to capture salient representations and variations within the data (Vincent et al., 2008; Rifai et al., 2011). Several variants of autoencoders (AEs) have been proposed in the literature, sometimes tailored for specific data modality such as text (Sutskever et al., 2014), images (Xu et al., 2014) and graph data such as molecules (Kusner et al., 2017).

*Variational autoencoders* (VAEs; Rezende et al., 2014; Kingma & Welling, 2014) have been introduced as probabilistic AEs whose encoders and decoders realize *stochastic* maps. Specifically, the encoder in a VAE draws embedding samples from a conditional distribution $\mathbf{z} \sim p(\mathbf{Z} \mid \mathbf{x})$, which is commonly realized as a simple distribution (most of the time, an isotropic Gaussian) parameterized by a neural backbone that receives $\mathbf{x}$ as input. Analogously, its decoder, draws reconstructions from a distribution $\hat{\mathbf{x}} \sim p(\mathbf{X} \mid \mathbf{z})$ conditioned on the embedding $\mathbf{z}$. Again, the distribution is assumed to have a simple form parameterized by a neural network (Ghosh et al., 2020). Neural parameterizations for both the encoder and decoder increase the expressiveness of these models. However, this increased expressiveness comes at the cost of reduced flexibility, especially when the models are required to perform advanced probabilistic inference tasks, such as computing marginals.

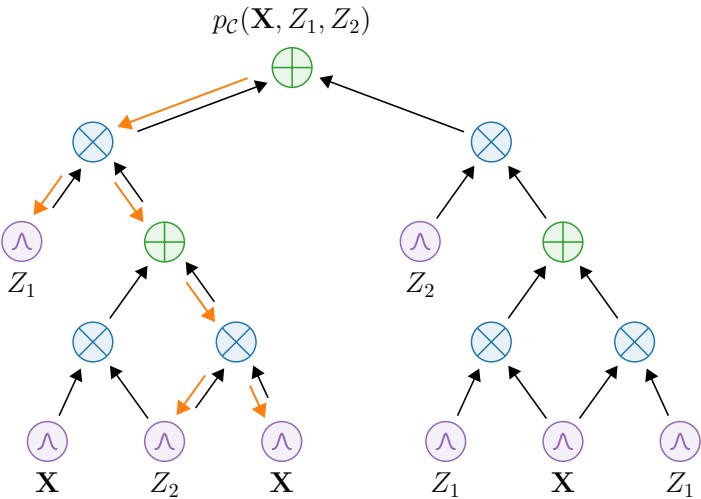

Figure 3: A PC over a set of data random variables $\mathbf{X}$ and two embedding variables $Z_1$ and $Z_2$. The black connections illustrate the flow of computation during a forward pass for inference. In contrast, the orange subgraph highlights a sampling-induced tree.

A classical example is that of missing input values (Collier et al., 2020; Williams et al., 2018). That is, when only some variables $\mathbf{X}_o \subset \mathbf{X}$ are observed, a neural parameterization of the encoder entails that computing $p(\mathbf{Z} \mid \mathbf{x}_o)$ is intractable, as it would require computing the marginal distribution $p(\mathbf{X}_o) = \int p(\mathbf{X}_o, \mathbf{X}_m) d\mathbf{X}_m$, where $\mathbf{X}_m = \mathbf{X} \setminus \mathbf{X}_o$ are the missing input features. As such, dealing with missing values for both AEs and VAEs is a challenging task, for which several heuristics and approximations have been proposed (Nazabal et al., 2020; Mattei & Frellsen, 2019; Simkus & Gutmann, 2023; 2024), see Section 4 for a detailed discussion.

**Probabilistic circuits.** A *probabilistic circuit* $\mathcal{C}$, is a computational graph representing a function $\mathcal{C}(\mathbf{X})$ over its inputs $\mathbf{X}$ (this is just a placeholder which can e.g. include data *and* embedding variables), where $\mathcal{C}(\mathbf{x}) \geq 0$ for any input $\mathbf{x}$, making $\mathcal{C}(\mathbf{X})$ an unnormalized distribution. The computational graph, parameterized by $\theta$, consists of three types of computational units: *input* $\wedge$, *sum* $\oplus$, and *product* $\otimes$. An input unit $c$ encapsulates a parameterized function $f_c(\mathbf{X}_c)$, with $\mathbf{X}_c \subseteq \mathbf{X}$ denoting its scope. The scopes for product and sum units, $\mathbf{X}_c$, are the union of their input units' scopes. Specifically, a product unit computes the product of its inputs $\prod_{d \in \text{in}(c)} d(\mathbf{X}_d)$, and a sum unit computes the weighted sum $\sum_{d \in \text{in}(c)} \theta_d^c d(\mathbf{X}_d)$. We assume sum unit weights to be normalized, i.e. $\sum_{d \in \text{in}(c)} \theta_d^c = 1$.

PCs can tractably (in time $\mathcal{O}(|\mathcal{C}|)$) and exactly compute evidence, marginal, conditional, and moment queries, which are essential to our proposed circuit-based encoding procedure, if it fulfills two structural properties: *smoothness* and *decomposability* (Darwiche & Marquis, 2002; Choi et al., 2020; Vergari et al., 2021).

**Structural properties for tractable marginals and sampling.** A sum unit $n$ is *smooth*, if all input units $c \in \text{in}(n)$ have the same scope, i.e., $\mathbf{X}_c = \mathbf{X}_n$. A product unit $n$ is *decomposable*, if all input units $c_i, c_j \in \text{in}(n)$ have pairwise disjoint scopes, i.e. $\mathbf{X}_{c_i} \cap \mathbf{X}_{c_j} = \emptyset$ for $c_i \neq c_j$. A PC is smooth and decomposable, if all of its sum units are smooth and all of its product units are decomposable. Fig. 3 presents an example of a smooth and decomposable PC. Smooth and decomposable PCs can be interpreted as hierarchical latent variable models, where a discrete latent variable $H_c$ is associated to each sum unit $c$. Such a latent variable has as many states as the number of input connections of its corresponding sum unit (Peharz et al., 2017). Under this light, the weights $\theta_d^c$ of a sum unit $c$ represent the probabilities of selecting a specific mixture component, i.e., one of its input connections. This latent variable interpretation enables exact and efficient ancestral sampling from the distribution $p_\mathcal{C}(\mathbf{X}, \mathbf{H})$, where $\mathbf{H}$ denote all the discrete latent variables associated to all sum units in the PC.[1]

---

[1]Usually, samples from $\mathbf{X}$ are retained and those from $\mathbf{H}$ discarded (Vergari et al., 2018).

Sampling proceeds as follows. From the output unit of a PC, one traverses the computational graph backwards, outputs before inputs, until the input distribution units are reached. This process constructs a *sampling-induced tree* in the PC. Fig. 3 illustrates this process. For a product unit, all input connections are followed, while for a sum unit, a single connection is sampled with probability proportional to its corresponding weight. Once an input distribution is reached, one can sample a feature corresponding to its scope according to the parametric distribution encoded in the unit. A complete sample $(\mathbf{x}, \mathbf{h})$ is retrieved by concatenating all features sampled from input units, and all latent variables $\mathbf{H}$ encountered in the backward traversal. However, this sampling procedure is inherently non-differentiable due to the discrete choices made at sum units. While methods to make PC sampling differentiable have been explored (Lang et al., 2022), their practical limitations, such as the poor sample quality and the gradient instability, motivate the need for more robust ways to sample PCs and integrate them in larger neural pipelines.

## 3 Autoencoding Probabilistic Circuits

We now introduce *autoencoding probabilistic circuits* (APCs), a framework to learn end-to-end representations in an autoencoding fashion using PCs, leveraging their tractable inference capabilities. Prior attempts to extract representations from PCs have explored two main avenues, notably within sum-product autoencoding (SPAE; Vergari et al., 2018). First, as PCs can be viewed as a specialized type of neural network, representations (termed ACT embeddings in SPAE) have been derived from the activations of selected circuit units. However, unlike typical neural networks, PCs are sparse computational graphs that often lack a clear, inherent notion of a "bottleneck" layer, making it challenging to identify and select which units provide compact and informative representations. Second, given their interpretation as hierarchical latent variable models with internal discrete latent variables (as discussed in Section 2), representations (termed CAT embeddings in SPAE) have been formed by inferring the maximum a posteriori assignments to these internal latent variables. Yet, modern PCs can involve an enormous number of such internal latent variables, making it non-trivial to select a concise and globally meaningful subset for representation. Crucially, these prior approaches primarily derive embeddings post-hoc, without explicitly integrating the learning of embeddings into the training process end-to-end.

In contrast, APCs introduce a fundamentally different approach by treating embeddings as first-class random variables within the PC itself. Instead of learning a separate conditional encoder as in VAEs (i.e., an inference model $p(\mathbf{Z} \mid \mathbf{x})$), we design a single PC, $\mathcal{C}$, that explicitly models the *joint probability distribution* $p_{\mathcal{C}}(\mathbf{X}, \mathbf{Z})$ over both data $\mathbf{X}$ and embeddings $\mathbf{Z}$. This is achieved by representing the embedding variables with their own dedicated, parametric input units within the circuit structure. The primary advantage of this formulation is that the encoder *is* the joint distribution itself, which remains fully tractable. Consequently, we can obtain embedding representations through exact probabilistic inference, such as by sampling from the true conditional posterior $\mathbf{z} \sim p_{\mathcal{C}}(\mathbf{Z} \mid \mathbf{x})$ *even for partially-observed inputs* $\mathbf{x}$ or performing maximum a posteriori (MAP) inference. This provides a more principled and direct foundation for probabilistic representation learning. Note that these *explicit* input embeddings $\mathbf{Z}$ are different from the *internal* discrete latent variables $\mathbf{H}$ of the PC (see previous section). While the latter are discrete latent variables, the former can be continuous.

### 3.1 Encoding with Tractable Joint Data-Embedding Distributions

As the encoder model, we construct a smooth and decomposable PC $\mathcal{C}$ that models the joint distribution $p_{\mathcal{C}}(\mathbf{X}, \mathbf{Z})$ of data and embedding random variables. The encoding process $f_{\mathcal{C}} : \mathbf{X} \mapsto \mathbf{Z}$ maps a data sample $\mathbf{x}$ to its embedding representation $\mathbf{z}$. We define the encoding $f_{\mathcal{C}}(\mathbf{x})$ as performing probabilistic state inference in the PC by conditionally sampling an embedding, given the data $\mathbf{x}$. In contrast to the VAE-based formulation, when $\mathcal{C}$ is a smooth and decomposable PC, we can perform arbitrary marginalization and conditioning, allowing us to directly sample embedding variables from the data-conditional distribution $p_{\mathcal{C}}(\mathbf{Z} \mid \mathbf{x})$. Conditional sampling in PCs consists of two passes through the network. First, a forward pass of the marginal $p_{\mathcal{C}}(\mathbf{x})$ for a given data sample $\mathbf{x}$ caches log-likelihoods at each input edge. Subsequently, a sampling pass with reweighted sum unit weights according to their cached log-likelihoods samples from the on $\mathbf{X} = \mathbf{x}$ conditioned marginal $p_{\mathcal{C}|_{\mathbf{x}=\mathbf{x}}}(\mathbf{Z}) = p_{\mathcal{C}}(\mathbf{Z} \mid \mathbf{X} = \mathbf{x})$. At test time we replace all sampling operations in the circuit with approximate

maximum a posteriori (MAP) inference.[2] Note that this is done with the purpose of enhancing the quality of the drawn samples and, while samples are not i.i.d. anymore, they are still valid for the purpose of generative modeling (or in our case, representation learning) and it is a common practice in the literature of VAEs, as discussed in Peharz et al. (2020a). We outline the full encoding algorithm in Appendix Algorithm 1.

Moreover, and in stark contrast to neural encoders, a PC-based encoding scheme allows us to infer the full embedding state $\mathbf{z}$ from *partial evidence*, where only a subset $\mathbf{X}_o \subseteq \mathbf{X}$ of the random variables are observed, and the rest is missing $\mathbf{X}_m = \mathbf{X} \setminus \mathbf{X}_o$. This is achieved through marginalizing the missing variables $\int p_{\mathcal{C}}(\mathbf{X}_o, \mathbf{X}_m)\, d\mathbf{X}_m$, without the need for any data imputation. On the other hand, neural networks assume all input values to be given. Consequently, data with missing observations must be manually preprocessed before being fed to a neural network. When not natively accounted for in a specific method, this preprocessing typically involves employing various imputation techniques such as mean imputation, multiple imputation (Rubin, 1989), or advanced methods like expectation-maximization algorithms and deep learning approaches (Yoon et al., 2018; Gondara & Wang, 2018; Luo et al., 2018). As a side effect, for neural networks these imputation methods usually introduce biases or assumptions that can affect model performance.

While not explicitly explored in this work, we emphasize that using a circuit encoder enables leveraging data-specific input or even embedding distributions to model random variables accurately. This is particularly beneficial in scenarios where we have prior knowledge about the specific data or embedding distributions or when each variable can be effectively represented as a mixture of multiple distributions of different types.

**Encoder Structure.**  A key design choice within the APCs framework is the architecture of the PC encoder. While our approach is agnostic to the specific topology beyond smoothness and decomposability, the chosen structure dictates how the joint distribution $p_{\mathcal{C}}(\mathbf{X}, \mathbf{Z})$ is decomposed and influences the learned representations. This architectural flexibility is both a strength and a challenge. While a similar large design space exists for neural networks, structural choices in PCs have direct and explicit semantic consequences. Specifically, the decomposition defined by product units encodes conditional independence assumptions directly into the model's architecture. This contrasts with neural networks where such relationships are learned implicitly. For APCs, this means the circuit structure determines the precise probabilistic relationship between data $\mathbf{X}$ and embeddings $\mathbf{Z}$, directly influencing the meaning of the learned representations. In our experiments, we employ architectures tailored to the data modality.

For tabular data, we use the architectural recipe of EinsumNetworks (Peharz et al., 2020a) with a RAT (Peharz et al., 2020b) structure. Here, inputs are modeled in random order in multiple random permutations without any quantification as to which split between data variables is a "good fit", given the data. Therefore, we simply choose to model embedding variables as input units alongside the data variables at the circuit's input layer. For image data, we construct a convolutional-style layerwise PC that builds a feature hierarchy through alternating product and sum layers. Each product layer reduces the height and width of the input by a factor of 2, building the product over all scopes in disjoint neighboring windows, similar to non-overlapping convolution kernels, when the stride is equal to the kernel size. After each product layer, a sum layer maps all input units of the same scope to a vector of sum units, similar to convolution in- and out-channels in traditional neural convolution layers. In this structure, each embedding input is randomly coupled with a random data variable unit via a local product unit. Further implementation details for these architectures are provided in Appendix B.1.

While these are simplistic choices for how we incorporate the embedding variables in our experiments, one could also move beyond the designs used here by leveraging structure learning algorithms or more principled heuristics (Loconte et al., 2025a). This could e.g. include the strategic placement of embedding variables at various depths within the circuit, allowing for hierarchical embeddings capable of distinguishing between low-level features and high-level abstractions.

---

[2]This is done by replacing sum units with max units and tracing backwards (from the circuit output to its inputs) the input configuration that (approximately) mostly activates the circuit (Mei et al., 2018; Vergari et al., 2021).

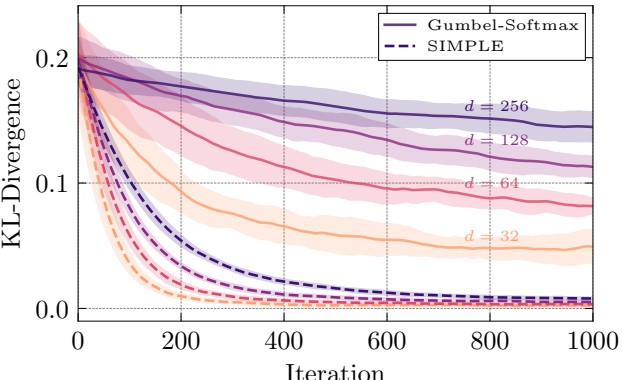

Figure 4: **SIMPLE outperforms Gumbel-Softmax in gradient estimation** as measured by the KL between ground-truth and learned sum units for different input dimensions $d$. SIMPLE converges faster and achieves a significantly lower KL-Divergence in a synthetic task to learn sum unit parameters using the mean squared error on samples drawn from a fixed ground-truth categorical distribution.

## 3.2 Differentiable Tractable Probabilistic Encoding

As discussed in Section 2, while exact (conditional) sampling from a smooth and decomposable PC is possible, it presents a challenge for gradient-based optimization: it is not differentiable. This non-differentiability arises from the inherent discreteness of the sampling procedure, specifically drawing from categorical distributions defined by sum unit weights and deterministically choosing successor units. Here, we make use of the concept of differentiable sampling for PCs introduced in Lang et al. (2022). This allows us to replace the discrete operations present in the sampling procedure of sum units with continuous, and thus differentiable, reparametrizations. We further improve the method proposed in Lang et al. (2022) and replace the Gumbel-Softmax trick with SIMPLE (Ahmed et al., 2023). SIMPLE is a gradient estimator for $k$-subset sampling that uses discrete sampling in the forward pass and exact conditional marginals in the backward pass. When we set $k = 1$, SIMPLE performs sampling from a mixture equal to what we require in sum units and can be used in-place of the Gumbel-Softmax trick. We outline how SIMPLE is applied to a sum unit in Appendix Algorithm 3. By incorporating SIMPLE into the differentiable sampling procedure for PCs, we can improve gradient estimation and overall performance of the autoencoding pipeline, overcoming the initial difficulties of differentiable sampling in PCs reported in Lang et al. (2022), which we show in Fig. 4. We designed a synthetic task where the objective is to learn sum unit parameters by differentiably sampling from this sum unit and minimizing the sample mean squared error to samples drawn from a fixed ground-truth categorical distribution. In this setting, SIMPLE consistently converged faster, achieved higher accuracy (as measured by the Kullback–Leibler divergence between the sum unit and the true categorical distribution), and exhibited lower variance compared to Gumbel-Softmax. These findings validate SIMPLE for more accurate and stable end-to-end training of APCs. We provide full details on this synthetic task in Appendix G.

## 3.3 Hybrid Decoding with Neural Networks

In APCs, while the encoding phase leverages the principled and tractable nature of PCs, the decoding process employs a neural network to reconstruct data from the learned embeddings. This design choice is motivated by the complementary strengths of neural networks, particularly their capacity for modeling complex, non-linear mappings. Therefore, in APCs, the decoder $g_\theta : \mathbf{Z} \mapsto \mathbf{X}$ is implemented as a separate neural network, distinct from the PC encoder $\mathcal{C}$. This neural decoder maps the embedding representation $\mathbf{z}$ back to the original data space $\mathbf{X}$, effectively complementing the probabilistic encoding with the modeling capacity and computational efficiency of neural networks during decoding. The specific architecture of the neural decoder can be tailored to the data type. For instance, CNNs are well-suited for image data, enabling the incorporation of implicit biases beneficial for capturing spatial relationships not explicitly modeled in the encoding circuit. Similarly, Transformer-based architectures can be leveraged for sequential or high-dimensional data due to their ability to model long-range dependencies and complex interactions. Beyond these, for achieving high-fidelity reconstructions and sample quality, advanced generative models such as diffusion models could also be integrated as decoders.

Importantly, this neural decoding strategy remains robust even when dealing with partial evidence. The tractable encoder $f_\mathcal{C}$'s ability to infer a complete embedding $\mathbf{z}$ from a partially observed input $\mathbf{x}_o$ ensures that the neural decoder always receives a full embedding vector. Consequently, the decoding process is unaffected by missing data in the input. We empirically explore the benefits of this hybrid approach by comparing APCs with neural decoders to APCs with self-decoding PCs in Section 5.6.

### 3.4 End-to-End Training of Tractable Encoder and Neural Decoder

APCs are trained end-to-end using objectives that encourage both accurate reconstruction and meaningful and robust embedding representations. We derive our training objective from a standard regularized autoencoding mechanism. The loss function is a sum of three crucial components: a reconstruction term, an embedding prior regularization term, and a joint data-embedding likelihood regularization term:

$$\mathcal{L} = \lambda_{\text{REC}} \cdot \mathcal{L}_{\text{REC}} + \lambda_{\text{KL}} \cdot \mathcal{L}_{\text{KL}} + \lambda_{\text{NLL}} \cdot \mathcal{L}_{\text{NLL}}. \tag{1}$$

The factors $\lambda$ allow us to assign specific weights to different objectives. Such weighting can, e.g., facilitate disentanglement (Higgins et al., 2017). A detailed investigation of alternative configurations for $\lambda$ is left to future work. For all experiments in Section 5, we set $\lambda_{\text{REC}} = \lambda_{\text{KL}} = \lambda_{\text{NLL}} = 1$, as this configuration already worked well in our scenario. In the following, we will describe each objective in detail.

**Reconstruction.** We use a reconstruction term, $\mathcal{L}_{\text{REC}}$, to ensure that embeddings represent sufficient information to reconstruct the original input data accurately. This objective, analogous to the one used in conventional autoencoders (Hinton & Salakhutdinov, 2006), encourages the model to learn a meaningful and compact embedding space that captures essential features of the data-generating distribution:

$$\mathcal{L}_{\text{REC}} = -\frac{1}{B} \sum_{i=1}^{B} \log p_\theta(\mathbf{x}_i \mid \mathbf{z}_i), \quad \text{where } \mathbf{z}_i \sim p_\mathcal{C}(\mathbf{Z} \mid \mathbf{x}_i), \tag{2}$$

where $B$ is the batch size, $\mathbf{x}_i$ is the $i$-th input sample.

**Embedding Regularization.** To ensure the learned embeddings are meaningful and to avoid overfitting, we regularize the embedding distribution to a chosen prior. While simple $L_p$ regularization could be applied to the embedding vectors $\mathbf{z}$, PCs allow for a more principled approach. Sampling in a PC instantiates the hidden sum unit variables $\mathbf{h} \in \mathbf{H}$ and induces a product unit tree with independent input units. This is highlighted as the orange subgraph in Fig. 3. Consequently, sampling embeddings from the conditional distribution $p_\mathcal{C}(\mathbf{Z} \mid \mathbf{x})$, given some data point $\mathbf{x}$, induces a tree $\mathcal{C}'$ in which embedding variables are mutually conditionally independent, given $\mathbf{x}$ and $\mathbf{h}$:

$$Z_j \perp Z_k \mid \mathbf{X} = \mathbf{x}, \mathbf{H} = \mathbf{h}, \quad \forall j, k \in 1, \dots, |\mathbf{Z}|, j \neq k. \tag{3}$$

This independence allows us to apply statistical distances between the induced embedding distribution and a chosen prior $q$. Specifically, we use the Kullback-Leibler divergence (KL):

$$\mathcal{L}_{\text{KL}} = \frac{1}{B} \sum_{i=1}^{B} \sum_{j=1}^{|\mathbf{Z}|} \text{KL}\big(p_{\mathcal{C}'}(Z_j \mid \mathbf{x}_i, \mathbf{h}_i) \,||\, q(Z_j)\big) \tag{4}$$

where $B$ is the batch size and $\mathbf{x}_i$ represents the $i$-th data sample. When choosing distributions of particular parametric forms, e.g. the exponential family, we can compute Eq. (4) analytically in closed form. Embedding distributions in $\mathcal{C}$ are chosen as Gaussian input units in all cases.

**Joint Data and Embedding Likelihood.** A key advantage of the PC encoder is its capacity to model the full joint distribution $p_\mathcal{C}(\mathbf{X}, \mathbf{Z})$. We leverage this by introducing a third loss component that maximizes the joint log-likelihood of the data and their inferred embeddings. This objective steers the encoder's parameters towards a maximum likelihood estimate (MLE) of the joint distribution, regularizing the reconstruction-focused training. Additionally, this helps in mitigating the well-known posterior collapse in which $p_\mathcal{C}(\mathbf{Z} \mid \mathbf{X})$

degenerates to $p_\mathcal{C}(\mathbf{Z})$, by enforcing a high likelihood on the joint distribution enforcing a direct relationship between $\mathbf{X}$ and $\mathbf{Z}$ in the encoder. Our ablation study in Section 5.6 empirically validates that this term is crucial for achieving high performance. We formalize this objective as the negative log-likelihood (NLL):

$$\mathcal{L}_{\text{NLL}} = -\frac{1}{B} \sum_{i=1}^{B} \log p_\mathcal{C}(\mathbf{x}_i, \mathbf{z}_i) \quad \text{where } \mathbf{z}_i \sim p_\mathcal{C}(\mathbf{Z} \mid \mathbf{x}_i). \tag{5}$$

**Training Via Knowledge Distillation.** While the APCs framework can be trained from scratch on raw data using the aforementioned objectives, its tractable marginals also enable data-free knowledge distillation from pre-trained generative latent variable models like VAEs. In this scenario, the APC model acts as a student, iteratively learning to capture the teacher's generative distribution without direct access to the original training data. The process involves sampling an embedding $\mathbf{z}$ from the APC's prior $p_\mathcal{C}(\mathbf{Z})$, which the teacher VAE's decoder $g_{\text{VAE}}(\mathbf{z})$ uses to generate synthetic data $\hat{\mathbf{x}}_{\text{VAE}}$. The corresponding embedding $\mathbf{z}_{\text{VAE}}$ is then obtained by encoding $\hat{\mathbf{x}}_{\text{VAE}}$ with the VAE's encoder $f_{\text{VAE}}(\hat{\mathbf{x}}_{\text{VAE}})$. For training the APC student, these synthetic data $\hat{\mathbf{x}}_{\text{VAE}}$ and teacher embeddigns $\mathbf{z}_{\text{VAE}}$ replace the "ground truth" data points $\mathbf{x}_i$ and their corresponding embeddings $\mathbf{z}_i$ in the loss terms. Specifically, we get

$$\mathcal{L}_{\text{KL,KD}} = \frac{1}{B} \sum_{i=1}^{B} \sum_{j=1}^{|\mathbf{Z}|} \text{KL}\big(p_{\mathcal{C}'}(Z_j \mid \mathbf{x}_{\text{VAE},i}, \mathbf{h}_i) \,\|\, q(Z_j)\big) \tag{6}$$

$$\mathcal{L}_{\text{NLL,KD}} = -\frac{1}{B} \sum_{i=1}^{B} \log p_\mathcal{C}(\mathbf{x}_{\text{VAE},i}, \mathbf{z}_{\text{VAE},i}) \tag{7}$$

$$\mathcal{L}_{\text{REC,KD}} = -\frac{1}{B} \sum_{i=1}^{B} \log p_\theta(\mathbf{x}_{\text{VAE},i} \mid \mathbf{z}_i), \quad \text{where } \mathbf{z}_i \sim p_\mathcal{C}(\mathbf{Z} \mid \mathbf{x}_{\text{VAE},i}) \quad . \tag{8}$$

A detailed data-free knowledge-distillation algorithm is outlined in Appendix Algorithm 2. While our main focus in Section 5 is on training APCs from scratch, we additionally explore the alternative perspective of training via data-free knowledge distillation from pretrained VAEs in Section 5.5.

## 4 Related Work

Having introduced APCs in the previous section, we now contextualize our framework by discussing prior work in representation learning using both PCs and neural networks and prior hybrid approaches between these two classes of models.

**Representation Learning with Probabilistic Circuits.** PCs have seen significant advancements in expressivity and learning algorithms including learning tensorized representations (Mari et al., 2023; Liu et al., 2023a;b; Ventola et al., 2023; Gala et al., 2024a;b; Loconte et al., 2025a), sparse structures (Di Mauro et al., 2015; 2017; Trapp et al., 2019; Vergari et al., 2019c; Dang et al., 2022a; Yang et al., 2023), and extending them to have negative parameters (Loconte et al., 2024; 2025b; Wang & Broeck, 2025). Nevertheless, their application to representation learning in an autoencoding fashion has been comparatively limited. The main work in this area is sum-product autoencoding (SPAE; Vergari et al., 2018), which proposed two encoding strategies for Sum-Product Networks: CAT embeddings, derived from MAP inference over categorical latent variables, and ACT embeddings, using circuit activations as continuous representations. However, SPAE derives these embeddings post-hoc from PCs trained for density estimation via maximum likelihood estimation. Crucially, there is no explicit loss function during SPAE's training that directly optimizes the quality of these embeddings, for example, based on their ability to generate accurate reconstructions. In Appendix D, we additionally evaluate the APC framework with an SPAE encoder and decoder and a neural decoder.

Other research has explored integrating PCs with deep generative models (DGMs). This includes using neural networks to guide PC structure learning or act as conditional components (Shih et al., 2021; Shao et al., 2022; Martires, 2024), incorporating deep learning inductive biases into PCs (Ventola et al., 2020; Butz et al., 2019; Yu et al., 2022), or using PCs within hybrid DGMs for tasks like knowledge distillation or

in combination with normalizing flows and VAEs (Liu et al., 2023a;b; Pevný et al., 2020; Sidheekh et al., 2023; Correia et al., 2023; Gala et al., 2024a). Dennis & Ventura (2017) proposed a hybrid model using two separate circuits for data and embeddings alongside a neural autoencoder to refine samples. In contrast to these approaches, APCs introduce a framework where probabilistic embeddings are *explicitly* modeled as random variables within a single PC encoder and are learned end-to-end.

**Neural Autoencoders for Representation Learning.** Neural autoencoders (AEs) form the foundation of many modern representation learning techniques (Hinton & Salakhutdinov, 2006). Early extensions like denoising autoencoders (DAEs; Vincent et al., 2008) and contractive autoencoders (CAEs; Salah et al., 2011) focused on learning robust and invariant features. The advent of variational autoencoders (VAEs; Kingma & Welling, 2014; Rezende et al., 2014) introduced a principled probabilistic approach, enabling generative modeling by learning a stochastic mapping to a latent space, typically regularized towards a simple prior. Despite their success, VAEs often rely on approximate inference and simplified posterior distributions (e.g., Gaussians), which can limit their expressiveness. Significant research has aimed to overcome these limitations through richer posterior families like normalizing flows (Rezende & Mohamed, 2015; Kingma et al., 2016; Louizos & Welling, 2016; Horvat & Pfister, 2021; Draxler et al., 2024), hierarchical latent structures (Sønderby et al., 2016; Vahdat & Kautz, 2020), discrete representations like VQ-VAEs (van den Oord et al., 2017; Razavi et al., 2019), and improved training objectives (Burda et al., 2016; Higgins et al., 2017; Tomczak & Welling, 2018). Deterministic variants like regularized autoencoders (RAEs; Ghosh et al., 2020) have also been explored to structure the latent space without the sampling complexities of VAEs, often relying on post-hoc density estimation for generation.

Furthermore, various neural AE adaptations have been proposed to address the real-world scenario of missing data. These include techniques such as importance-weighted training (MIWAE, Mattei & Frellsen, 2019; not-MIWAE, Ipsen et al., 2021; supMIWAE, Ipsen et al., 2022), tailored likelihoods for heterogeneous missing data (HI-VAE; Nazabal et al., 2020), modified losses in DAEs (mDAE; Dupuy et al., 2024), and advanced inference or sampling strategies in hierarchical or iterative models (Peis et al., 2022; Kuang et al., 2024). While powerful, these neural autoencoding paradigms, even those specialized for missing data, often involve approximate inference for their probabilistic embeddings or necessitate imputation strategies for incomplete data. Another self-supervised paradigm is the masked autoencoder (MAE; He et al., 2022), which is a vision transformer-based (ViT; Dosovitskiy et al., 2020) approach that learns representations by encoding only a selected subset of visible input patches to reconstruct the masked ones. MAEs are trained on specifically chosen patch masking patterns and map each unmasked input patch to its own patch-embedding. This is distinct from models like VAEs and APCs, which generate a single embedding for the entire input. A related category is the neural process (NP) family (Garnelo et al., 2018b;a; Kim et al., 2019; Gordon et al., 2020). NPs learn distributions over functions by conditioning on a set of observed context points to make predictions at arbitrary target locations. Their focus on learning a function-level representation from a context set distinguishes them from VAEs and APCs, which aim to learn a single, holistic embedding for an entire data instance. In contrast to these fully neural based models, APCs leverage a PC encoder for exact and tractable probabilistic inference over embeddings and the capability to handle missing observations via marginalization. The empirical comparison in the following section will further substantiate these distinctions.

## 5 Empirical Evaluation

To measure the effectiveness of APCs, we conduct a comprehensive series of experiments designed to evaluate their performance and robustness across multiple tests against an array of model choices. Our evaluation aims to answer the following four key research questions. **RQ1** (Section 5.1): How effectively can APCs reconstruct data, particularly in the presence of missing values, compared to established PC-based and neural approaches? **RQ2** (Sections 5.2 and 5.3): Do the explicit probabilistic embeddings learned by APCs yield high-quality, informative representations that are beneficial for downstream classification tasks, especially when inputs are incomplete? **RQ3** (Section 5.4): Do modern autoencoder architectures, training procedures, and higher model capacities already improve robustness against missing data, or does this remain a challenge? **RQ4** (Section 5.5): Can APCs act as effective student models for data-free knowledge distillation from pre-trained generative latent variable models and improve robustness to missing data compared to the teacher?

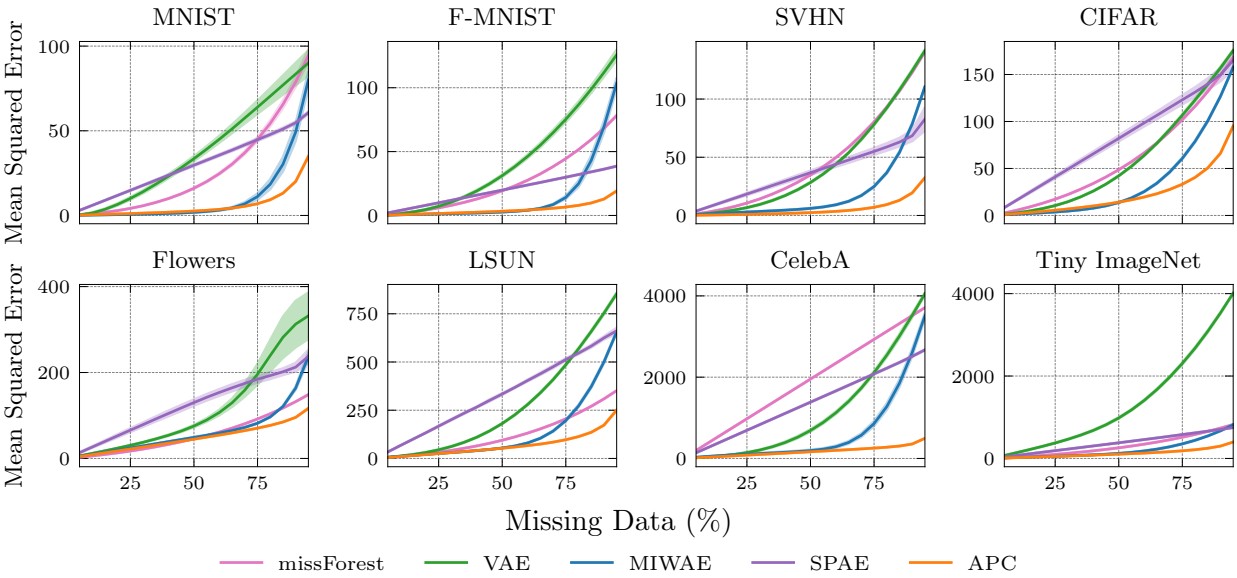

Figure 5: **APCs can deliver lower reconstruction errors than PC, VAE, and missForest baselines in the asymptotic regime of MCAR-style randomly missing data**. Reconstruction mean squared error is reported across all image datasets as the degree of MCAR corruption increases from 0% to 95%.

To address these questions, we include the existing autoencoding-mechanism SPAE introduced in Vergari et al. (2018) as a PC-based reference, a vanilla variational autoencoder (Kingma & Welling, 2014) as a neural autoencoding baseline, MIWAE (Mattei & Frellsen, 2019) as a VAE variant specifically designed for handling missing data through multiple imputation rounds, and missForest (Stekhoven & Bühlmann, 2011), an iterative imputation method based on random forests as a strong classic machine learning baseline. All models are evaluated on common image benchmark datasets, as well as the 20 DEBD binary tabular datasets (Lowd & Davis, 2010; Haaren & Davis, 2012; Bekker et al., 2015; Larochelle & Murray, 2011).

## 5.1 Reconstructions: Comparing Circuit and Neural Encoders

We assess reconstruction robustness against missing data by evaluating models under two primary missing data mechanisms: missing completely at random (MCAR) and missing at random (MAR). For MCAR, we introduce increasing levels of input corruption by randomly dropping pixels chosen from a uniform distribution, with the percentage of missing data ranging from 0% to 95% in 5% steps. This is the default corruption method unless otherwise specified. For MAR, we evaluate specific structured missingness patterns, such as missing entire regions of an image (e.g., left-to-right, center-to-border); one could also argue that rather than MAR, this is a data-type specific structured version of MCAR. All models, with the exception of MIWAE, were trained on the complete datasets without missing values. For MIWAE, training was performed using MCAR-style data in which 50% of entries were missing. For each metric, we report the missing-input MSE averaged over the different levels of corruptions. We repeat all experiments five times with different seeds and report their mean and standard deviation. We refer the reader to Appendix B for additional details. Notably, missForest is an imputation algorithm that operates directly in the input space, unlike the autoencoding methods, which learn a compressed representation in an embedding space. Consequently, when no data is missing (0% corruption), missForest can achieve a perfect reconstruction error of 0.0. In contrast, autoencoder-based approaches are inherently limited by their embedding bottleneck, resulting in non-zero reconstruction error even with complete inputs.

We first evaluate the reconstruction capabilities of APCs and examine performance on image data. In Fig. 5, we show MSE reconstruction error trends (SSIM in Appendix Figure 12) as the percentage of MCAR-style missing pixels increases from 0% to 95%. While all models perform comparably at low corruption levels, their

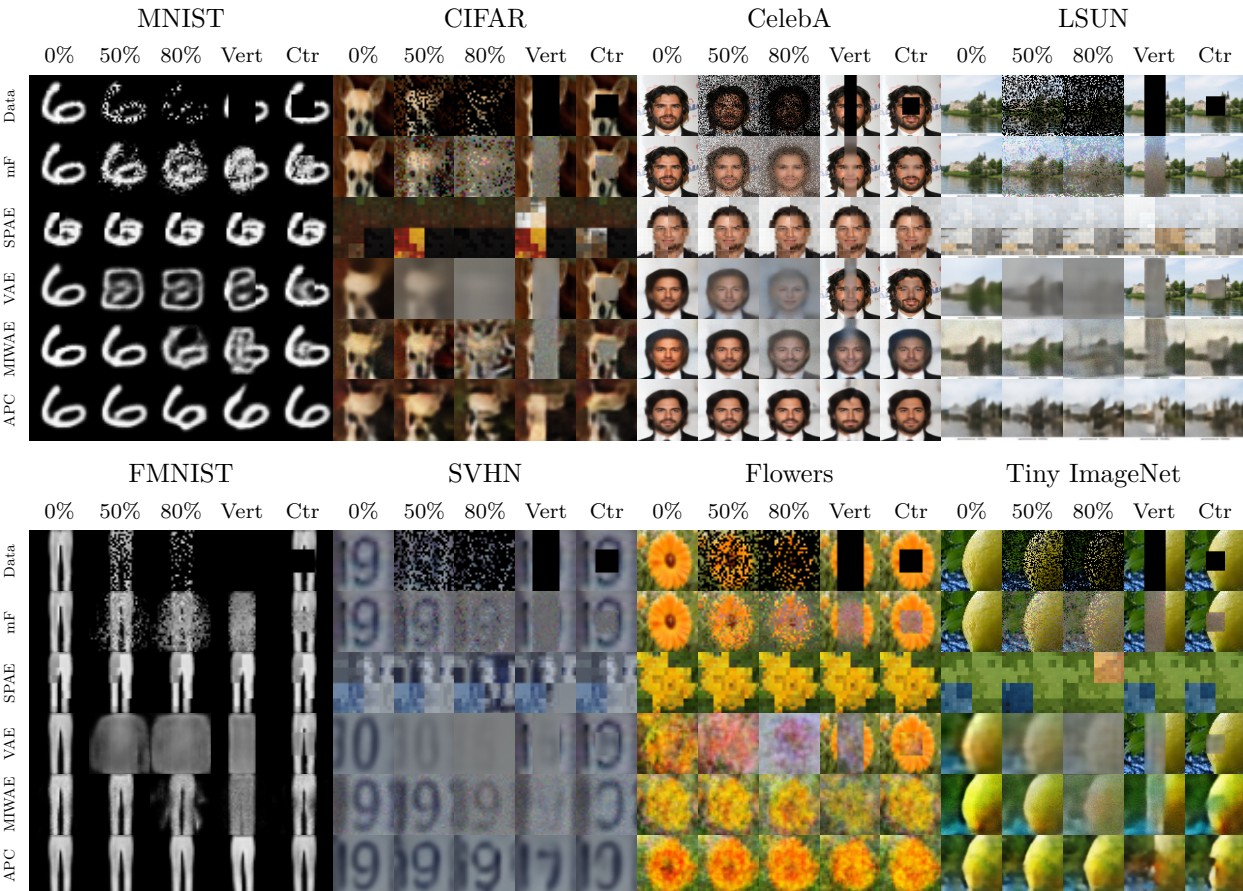

Figure 6: **While neural encoder models quickly collapse with an increase in missing data ratio and mostly fail on MAR corruptions, APCs are able to maintain a stable reconstruction, even at high degrees of MAR and MCAR corruption.** The first row (Data) shows inputs with different corruptions, and subsequent rows represent the respective model reconstructions.

results diverge quickly as corruption increases: neural autoencoders show rapid performance degradation. At the same time, APCs maintain lower reconstruction error even at high corruption levels. It is important to highlight that APC, VAE, and MIWAE utilize *the same neural decoder architecture*. Therefore, the observed differences in performance can be solely attributed to the choice of the encoder model and the corresponding encoding scheme. APCs demonstrate superior performance across all datasets, with lower MSE and higher SSIM compared to both neural and PC-based models. As an aggregate, we additionally provide the average reconstruction error over all corruption levels for MSE in Appendix Table 2 and for SSIM in Appendix Table 5.

In addition, qualitative examples of reconstructions with missing data in Fig. 6 visually confirm our quantitative findings. We show reconstructions for 0%, 50%, and 80% MCAR-style missing data and vertical-band as well as center-square MAR-style missing data. As missing data increases, neural autoencoders produce increasingly blurry and eventually unrecognizable reconstructions. In contrast, APCs maintain recognizable reconstructions even at high levels of missing data or images missing complete patches, preserving key structural elements and details across all datasets. We attribute this robustness to the encoder circuit's ability to handle partial observations through tractable marginalization natively.

While image data inherently contains spatial structure and correlations that neural architectures can exploit through layers with implicit biases such as convolutions, tabular data typically lacks such inherent structure. We therefore additionally investigate whether APCs' strong performance on image datasets generalizes to

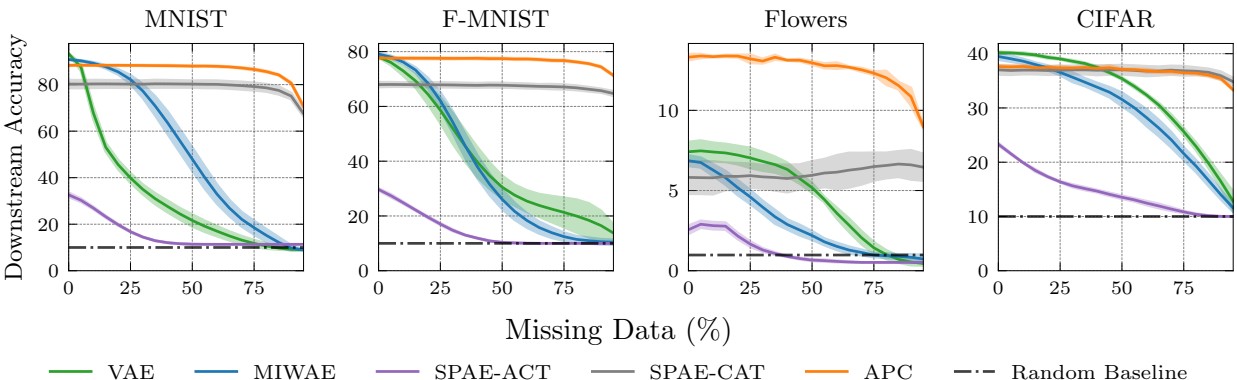

Figure 7: **APCs produce linearly seperable embeddings better than VAEs.** Downstream task accuracy using a Logistic Regression model under different MCAR-style corruption levels (0% to 95%) for MNIST, F-MNIST, Flowers, and CIFAR. We observe that APCs generate embeddings even under high corruption levels that are still linearly separable. Neural encoders (VAE/MIWAE), on the other hand, quickly collapse their embedding representation and thus drop in downstream task performance, when data is missing.

tabular data. Appendix Table 3 presents reconstruction performance on the 20 DEBD binary tabular datasets under increasing percentages of missing values. APCs outperform all other models on 18 out of 20 datasets, demonstrating that their advantages are not limited to image data but generalize to tabular datasets. This is particularly significant for tabular data applications where missing values are common in real-world scenarios, such as medical records, survey and financial data, or environmental modeling.

## 5.2 Embedding Quality and Downstream Task Performance

While quantitative and qualitative experiments in Section 5.1 so far have investigated reconstruction performance, a key indicator of representation learning quality is how well the embeddings can be leveraged for downstream task applications after *unsupervised pre-training* of the autoencoding model. Following a common practice in representation learning for evaluating the quality of unsupervisedly learned features (Bengio et al., 2012; Alain & Bengio, 2017), we assess the utility of our models' embeddings by treating them as fixed inputs for a supervised classifier. For this, a logistic regression model was trained iteratively using stochastic gradient descent (SGD) on these extracted embeddings. We specifically use a logistic regression model to evaluate the learned embeddings' ability to capture independent and useful data representations. For further details on the training protocol of this downstream task, we refer the reader to Appendix B.

The downstream task performance is measured by the downstream task classification accuracy (DS-Acc. ↑), providing insights into the quality and linear separability of the learned representations. Fig. 7 presents downstream classification accuracy of the logistic regression model trained on embeddings extracted from four image datasets under increasing levels of MCAR-style corruption. The results reveal a stark contrast between APCs and neural autoencoder variants: while neural encoders achieve slightly higher accuracy with complete data (0% missing), APC embeddings maintain their classification performance even as corruption increases to severe levels (90%), whereas neural encoder embeddings show immediate degradation in downstream performance, quickly converging towards a baseline random guessing estimator, when data is missing. This pattern is consistent across all datasets. The robustness to increasing proportions of missing data can again be attributed to the circuit's fundamental capability for tractable marginalization, allowing inference of embeddings even with incomplete observations without additional imputation methods.

To provide visual insight into these performance differences, Fig. 8 shows t-SNE projections of embedding spaces for the MNIST test dataset split at different corruption levels. Because SPAE-ACT embeddings represent circuit activations, which are log-likelihoods and can have large negative values, we normalize them before t-SNE to ensure a comparable input range with other models. With complete data (0% corruption), both neural autoencoders and APCs produce well-structured embeddings with clear class

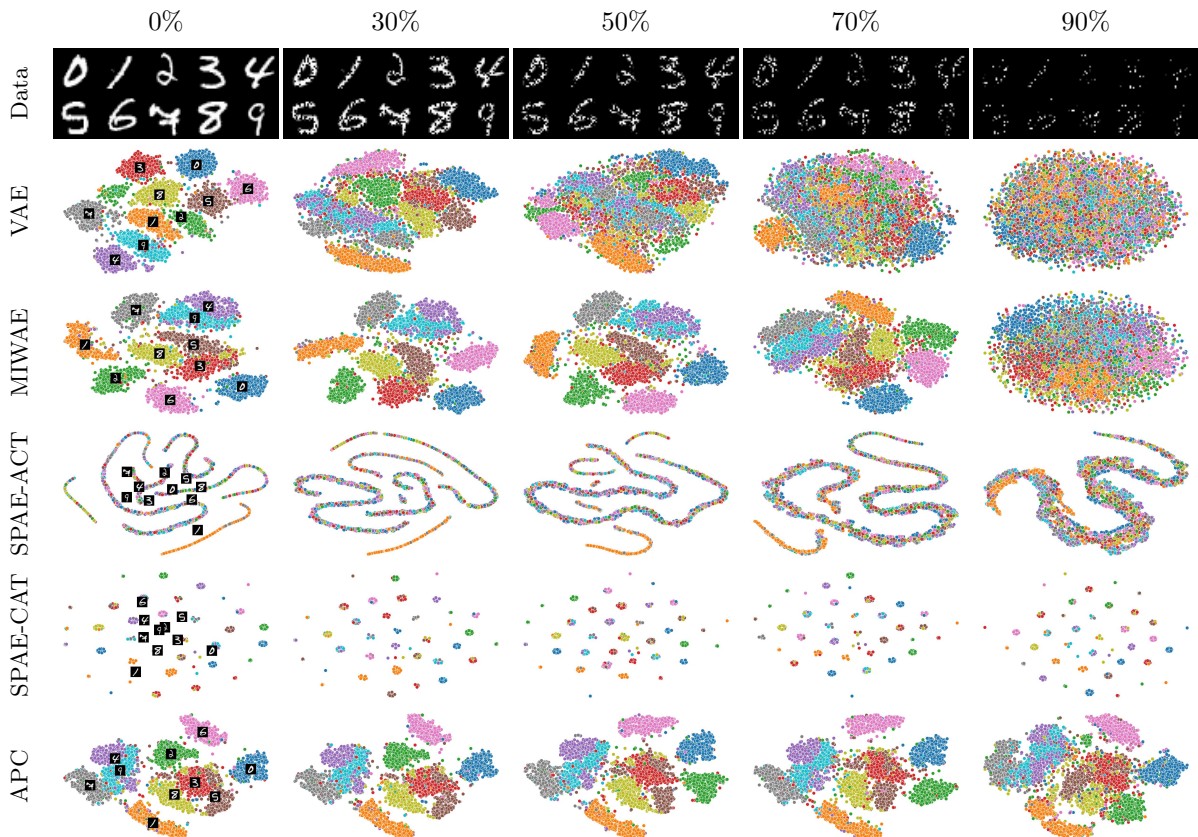

Figure 8: **APCs keep a stable embedding space even when corruption is high**. T-SNE projections of MNIST model embeddings for different corruption levels of missing data. Each dot is an MNIST test data point and colored according to its class. While neural encoders (*AE) initially (0%) can separate different classes in the embedding space, their representation starts to heavily degrade with increasing corruption levels until no separation is possible.

separation. However, as corruption increases, neural encoder, even MIWAE, which is specifically trained on missing data, embeddings progressively collapse into an unstructured mass with no class separation. In contrast, APC embeddings maintain their structure with distinct class clusters even at 90% corruption, visually confirming our quantitative results of Fig. 7 and their robustness to missing data.

## 5.3  Embedding Space Analysis

A key advantage of APCs is the fact that the encoder models the joint data-embedding distribution $p_{\mathcal{C}}(\mathbf{X}, \mathbf{Z})$ explicitly and can tractably compute marginals, enabling additional probabilistic capabilities compared to standard and variational autoencoders. In this section, we briefly examine APCs from a probabilistic perspective by investigating the embedding space and samples decoded from the embedding space.

Unlike VAEs, which implicitly model the approximate posterior distribution $q(\mathbf{Z} \mid \mathbf{X})$ using a neural network and rely on approximate inference, APCs directly model the joint distribution $p_{\mathcal{C}}(\mathbf{X}, \mathbf{Z})$ and allow for exact and tractable probabilistic queries. VAEs are trained to encourage the encoder's approximate posterior $q(\mathbf{Z} \mid \mathbf{X})$ to approach a prior distribution, typically a standard normal distribution $\mathcal{N}(0, 1)$. This is achieved by incorporating a KL term in the loss function that measures the discrepancy between $q(\mathbf{Z} \mid \mathbf{X})$ and $\mathcal{N}(0, 1)$. Samples are drawn from $\mathcal{N}(0, 1)$ after training, assuming the learned approximate posterior is a good approximation of the true posterior $p(\mathbf{Z} \mid \mathbf{X})$. However, the encoder rarely perfectly matches the true posterior, leading to a mismatch between the true posterior and the distribution used for sampling. In contrast, APCs allow us to sample embeddings directly from the exact marginal distribution $\mathbf{z} \sim p_{\mathcal{C}}(\mathbf{Z})$, obtained tractably

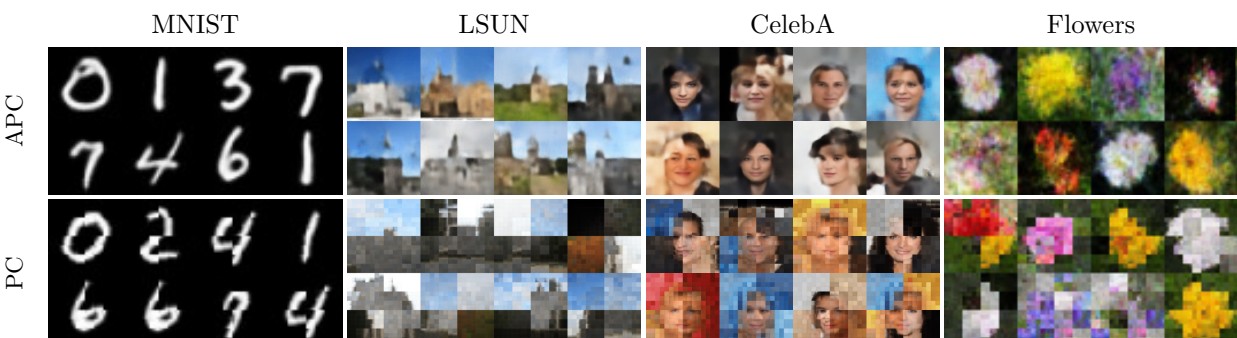

Figure 9: **APCs circumvent known visual sampling artifacts of traditional PCs**. APC and vanilla PC samples from models trained on MNIST, LSUN (Church), CelebA, and Flowers. APCs successfully learn the data-generating distribution and can produce novel samples.

through marginalization of the PC: $\int p_{\mathcal{C}}(\mathbf{X}, \mathbf{Z}) \, d\mathbf{X}$. These embeddings are then decoded to generate new data samples. Fig. 9 showcases the decoded embedding samples across datasets of varying complexity and contrasts them with a PC solely trained with MLE. The generated samples exhibit diversity and maintain the structural characteristics of their respective datasets, ranging from clear digit forms in MNIST to more complex architectural features in LSUN and facial attributes in CelebA. This is additionally confirmed by our quantitative evaluation of FID and KID scores in Appendix Table 10. Vanilla PC samples exhibit well-known visual artifacts related to the circuit structure. APCs circumvent this by deferring the decoding to a neural network. This demonstrates that APCs successfully capture the underlying data distribution through their tractable encoder, generating higher quality samples by decoding from the embedding distribution and natively avoiding posterior approximation discrepancies inherent in VAEs due to the intractable nature of their posterior distributions.

Furthermore, Tomczak & Welling (2018) have shown that the optimal prior $p^*(\mathbf{z})$ after fitting a VAE is the aggregate variational posterior $\frac{1}{N} \sum_{i=1}^{N} q_\theta(\mathbf{z} \mid \mathbf{x}_i)$. The discrepancy between this complex multimodal distribution and the simple unimodal prior typically used for sampling leads to the "prior hole problem" (Hoffman & Johnson, 2016; Aneja et al., 2020), where the prior assigns high probability to regions of the embedding space that are not covered by the aggregate posterior. Decoding samples from these "holes" often results in poor quality generations. To mitigate this, standard approaches often resort to a two-stage process: training the VAE first, and subsequently fitting a separate density estimator (e.g., a GMM or another VAE) to the aggregate posterior samples post-hoc (Ghosh et al., 2020; Dai & Wipf, 2019). APCs circumvent this issue entirely without requiring a second training stage. The marginal distribution $p_{\mathcal{C}}(\mathbf{Z})$ of the encoder can be viewed as a direct approximation of this aggregate posterior. Since APCs model the joint distribution, this marginal naturally covers the regions of the embedding space occupied by the data.

### 5.4 Model Capacity Is Insufficient for Missing Data Robustness Even in State-Of-The-Art VAEs

Our preceding analyses have primarily focused on neural encoder and decoder architectures. However, the autoencoding, particularly the VAE landscape, has witnessed significant advancements, leading to more sophisticated and complex architectures and training procedures. Notable examples include Vector Quantized VAEs (VQ-VAE) (van den Oord et al., 2017) and VQ-VAE2 (Razavi et al., 2019), as well as very deep models like VDVAE (Child, 2020) and hierarchical ones like NVAE (Vahdat & Kautz, 2020). This progress raises a critical question: does the sheer scale and complexity of modern VAEs inherently grant them robustness against missing data? While specialized methods and training recipes exist for this problem, our goal here is not to make a conclusive claim about the general capabilities of all modern VAEs. Instead, we aim to preliminarily investigate whether scaling up a VAE architecture is in itself sufficient to overcome the challenges posed by incomplete inputs. We therefore evaluate NVAE under MAR and MCAR corruptions as a representative for modern, high-capacity VAE architectures.

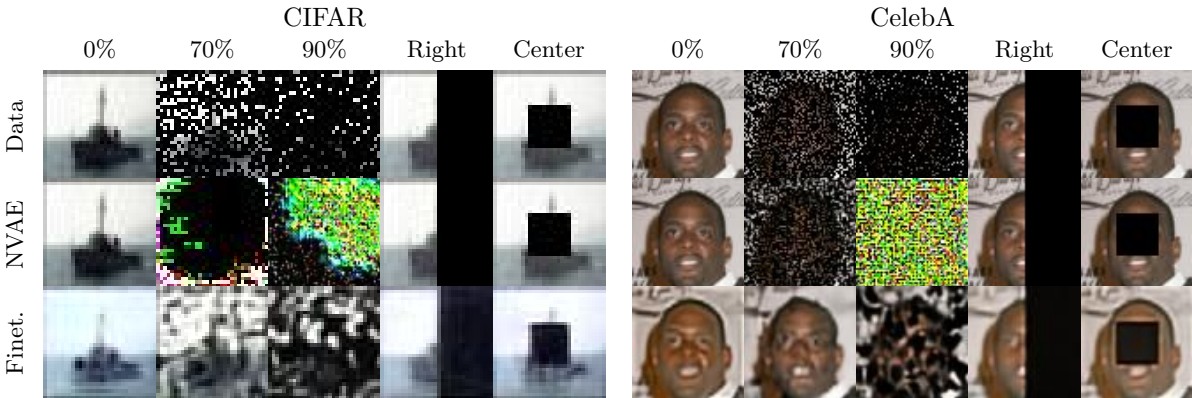

Figure 10: **Even modern VAEs cannot natively handle data corruption robustly.** Example reconstructions from NVAE and MCAR-finetuned NVAE on CIFAR and CelebA samples with MCAR/MAR corruption. Standard NVAE retains zero-imputed values, collapsing at high corruption (e.g., 90%). MCAR-finetuning offers marginal improvement for MCAR patterns but fails to generalize to other MAR patterns (e.g., missing right half or square in the center).

We employed the official publicly available pretrained NVAE models for CIFAR and CelebA from Vahdat & Kautz (2020). These models were evaluated on image reconstruction tasks with varying levels of data missingness: 0%, 70%, and 90% using MCAR-style patterns and two MAR-style patterns (missing right-half, missing center-square), as illustrated in Fig. 10. Initial experiments revealed that the base NVAE model tended to either keep the zero-imputed values or exhibit failure modes such as oversaturation, which were not seen in previous VAE experiments. To address this, we finetuned each NVAE model (labeled "Finet." in Fig. 10) on its respective dataset with MCAR patterns of $p = 50\%$ uniformly missing pixels for an additional 25% of its original training epochs. This finetuning process requires careful calibration, as finetuning degrades the model's original reconstruction capabilities or can lead to complete collapse. As depicted in Fig. 10, the base NVAE achieves nearly perfect reconstructions on complete data, whereas the finetuned model exhibits significantly poorer performance. This observation is supported by a substantial increase in bits-per-dimension (BPD) on the full test sets: from 2.91 to 16.93 for CIFAR and from 2.27 to 104.26 for CelebA, for the base to finetuned models, respectively. Although the finetuned model demonstrates some ability to reconstruct images with 70% missing pixels, it remains susceptible to similar failure modes as the base model at 90% missingness and cannot reconstruct MAR-style corruptions at all.

## 5.5 Application: Data-Free Knowledge Distillation

Pre-trained autoencoding models are increasingly shared online and offer powerful generative capabilities. However, a significant limitation, as demonstrated in our earlier evaluations (Sections 5.1 to 5.3), is their often fragile performance when confronted with MCAR and MAR corrupted input data. This raises a practical question: can we transfer the generative knowledge of a pretrained VAE to our APC framework when the original training data is unavailable? Knowledge distillation (KD) has emerged as a powerful technique for model compression, adaptation, and continual learning by transferring knowledge from a teacher model to a student model (Hinton et al., 2015). KD has seen an increasing amount of application in natural language processing (Tang et al., 2019; Jiao et al., 2020; Mou et al., 2016), computer vision (Liu et al., 2017; Zhou et al., 2018; Yim et al., 2017), and speech recognition (Hinton et al., 2015; Lu et al., 2017; Ramsay et al., 2019). Traditionally, this process requires access to the original training data, which may not always be available due to privacy concerns, proprietary restrictions, or storage constraints. Data-free knowledge distillation seeks to overcome these challenges by extracting transferable knowledge solely from the teacher's learned representations without requiring access to original data. Approaches range from generative adversarial network (Goodfellow et al., 2014) based and inspired methods (Chen et al., 2019; Han et al., 2021) to those that first generate synthetic data through specific distillation objectives and then train students on this

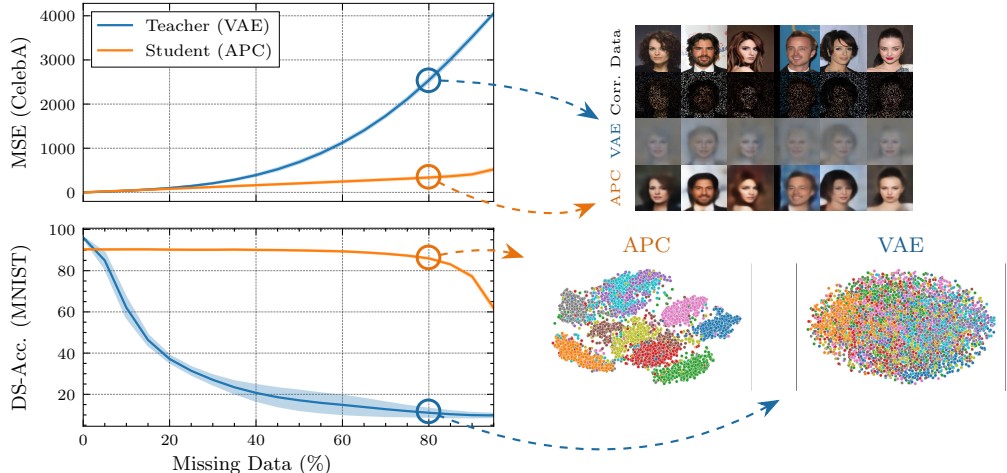

Figure 11: **APCs are robust data-free knowledge distillers.** Data-free Knowledge Distillation from VAE to APC. The distilled APC model can learn the VAE's distribution and improves in reconstruction and downstream performance under data corruptions.

synthetic data (Mordvintsev et al., 2015; Yin et al., 2020; Braun et al., 2023), align intermediate features between teacher and student (Wang, 2021a) to black-box hard-label based methods (Wang, 2021b).

In this section, we explore the capabilities of APCs as robust data-free knowledge distillers. Given the explicit probabilistic nature and tractable inference in APCs, we investigate their potential as student models to distill knowledge from generative latent variable models, such as VAEs, while circumventing the need for the original dataset. This approach allows us to examine whether APCs can learn the data distribution captured by a VAE teacher, while additionally benefiting from APCs robustness shown in the previous sections. We outline the data-free knowledge distillation procedure in Appendix Algorithm 2. In short, we sample an embedding from the APC prior $p_{\mathcal{C}}(\mathbf{Z})$ which is used by the teacher VAE to generate synthetic data samples. We then train the APC according to the same objectives outlined in Section 3 by treating the synthetic samples as reconstruction ground truth.

In Fig. 11, we show the results of knowledge distillation from a VAE to an APC of similar capacity and the same decoder architecture illustratively on two example datasets. The top graph quantifies reconstruction error on CelebA, measured as the MSE, as a function of MCAR-style missing data percentage. The teacher VAE exhibits a sharp increase in MSE as data corruption intensifies. In contrast, the student APC maintains lower reconstruction error, demonstrating its effectiveness in knowledge distillation and showing the same characteristics of APCs observed in the previous sections. In addition, we highlight the qualitative results of reconstructions with 80% randomly missing data on the right, where the APC student model more faithfully reconstructs corrupted images compared to its teacher VAE model.

The bottom graph evaluates downstream task performance on MNIST, measured by logistic regression classification accuracy using embeddings obtained from the models. As missing data increases, as observed earlier, the teacher VAE's performance rapidly deteriorates, whereas the student APC maintains high accuracy, indicating superior robustness of its learned embeddings towards corruptions. We once more highlight these results qualitatively on the right as t-SNE visualizations of the embedding spaces at 80% missing data, which further reinforces our findings: While the teacher VAE embeddings collapse into a dispersed and unstructured distribution, the student APC embeddings maintain well-separated clusters, suggesting better downstream utility. To provide further evidence, we repeat these experiments for every image and tabular dataset explored in Section 5 and show results for teacher and student reconstructions and downstream task accuracy under full evidence (Full Evi.) and average MSE over an increasing level of corruption (0% to 95% in 5% steps) in Appendix Table 8. The results confirm that APCs closely match the VAE's performance under full evidence while surpassing it in robustness to missing data, achieving lower reconstruction errors and higher downstream

Table 1: **Every APC component makes a distinct and complementary contribution to the framework.** Ablation study examining the contribution of key APC components: differentiable sampling for end-to-end training (Diff.), neural decoder (NN-Dec.), KL embedding regularization ($\mathbf{Z}$-Reg.), and joint data-embedding log-likelihood regularization ($p(\mathbf{x}, \mathbf{z})$). Performance is evaluated using MSE ($\downarrow$), downstream classification accuracy (DS-Acc. $\uparrow$) under full evidence conditions (Full Evi.), and average MSE metrics measuring robustness.

| | Diff. | NN-Dec. | $\mathbf{Z}$-Reg. | $p(\mathbf{x}, \mathbf{z})$ | MSE ($\downarrow$) | | DS-Acc ($\uparrow$) | |
| --- | --- | --- | --- | --- | --- | --- | --- | --- |
| | | | | | Full Evi. | MCAR | Full Evi. | MCAR |
| **MNIST** | ✓ | ✗ | ✗ | ✗ | $79.98$ $_{\pm 10.3}$ | $63.84$ $_{\pm 7.36}$ | $69.28$ $_{\pm 2.69}$ | $39.16$ $_{\pm 5.43}$ |
| | ✗ | ✓ | ✓ | ✓ | $55.21$ $_{\pm 1.24}$ | $26.57$ $_{\pm 0.48}$ | $59.69$ $_{\pm 3.05}$ | $58.01$ $_{\pm 3.11}$ |
| | ✓ | ✗ | ✓ | ✓ | $20.73$ $_{\pm 0.66}$ | $17.12$ $_{\pm 0.30}$ | $50.70$ $_{\pm 3.50}$ | $45.79$ $_{\pm 3.24}$ |
| | ✓ | ✓ | ✗ | ✓ | $8.18$ $_{\pm 0.07}$ | $8.43$ $_{\pm 0.02}$ | $88.13$ $_{\pm 0.22}$ | $82.94$ $_{\pm 0.19}$ |
| | ✓ | ✓ | ✓ | ✗ | $7.45$ $_{\pm 0.19}$ | $19.87$ $_{\pm 1.14}$ | $85.98$ $_{\pm 1.77}$ | $65.58$ $_{\pm 3.65}$ |
| | (✓) | ✓ | ✓ | ✓ | $7.60$ $_{\pm 0.17}$ | $10.07$ $_{\pm 0.21}$ | $86.62$ $_{\pm 0.29}$ | $81.97$ $_{\pm 0.63}$ |
| | ✓ | ✓ | ✓ | ✓ | $\mathbf{4.13}$ $_{\pm 0.10}$ | $\mathbf{5.06}$ $_{\pm 0.05}$ | $\mathbf{88.32}$ $_{\pm 0.14}$ | $\mathbf{86.85}$ $_{\pm 0.11}$ |
| **CIFAR** | ✓ | ✗ | ✗ | ✗ | $441.32$ $_{\pm 17.5}$ | $291.15$ $_{\pm 7.43}$ | $34.76$ $_{\pm 0.51}$ | $24.32$ $_{\pm 0.51}$ |
| | ✗ | ✓ | ✓ | ✓ | $193.77$ $_{\pm 0.00}$ | $92.07$ $_{\pm 0.00}$ | $29.80$ $_{\pm 0.00}$ | $29.51$ $_{\pm 0.00}$ |
| | ✓ | ✗ | ✓ | ✓ | $60.55$ $_{\pm 0.86}$ | $68.31$ $_{\pm 0.93}$ | $33.09$ $_{\pm 0.58}$ | $29.24$ $_{\pm 0.64}$ |
| | ✓ | ✓ | ✗ | ✓ | $25.45$ $_{\pm 0.20}$ | $22.52$ $_{\pm 0.11}$ | $36.39$ $_{\pm 0.44}$ | $31.11$ $_{\pm 0.15}$ |
| | ✓ | ✓ | ✓ | ✗ | $35.70$ $_{\pm 0.13}$ | $67.52$ $_{\pm 0.56}$ | $33.71$ $_{\pm 0.27}$ | $25.79$ $_{\pm 0.50}$ |
| | (✓) | ✓ | ✓ | ✓ | $17.04$ $_{\pm 0.19}$ | $28.23$ $_{\pm 0.24}$ | $35.47$ $_{\pm 0.23}$ | $34.13$ $_{\pm 0.09}$ |
| | ✓ | ✓ | ✓ | ✓ | $\mathbf{16.29}$ $_{\pm 0.14}$ | $\mathbf{20.64}$ $_{\pm 0.08}$ | $\mathbf{37.61}$ $_{\pm 0.27}$ | $\mathbf{36.90}$ $_{\pm 0.14}$ |

accuracy across most datasets. This highlights APCs' ability to effectively distill knowledge from VAEs without access to original training data while enhancing robustness to data corruption.

## 5.6 Ablation Studies

Following an extensive evaluation of APCs across a diverse range of datasets and against circuit-based, neural, and classic models in the previous sections, we now conduct an ablation study to isolate and quantify the contribution of each component within our framework, thereby strengthening the understanding of APCs. As shown in Table 1, we analyze both the removal of all components simultaneously and the exclusion of individual components while maintaining others. We evaluate four key architectural elements: (1) differentiable sampling (Diff.), which enables end-to-end gradient flow between the encoder circuit and decoder network; (2) the neural decoder (NN-Dec.), as opposed to using the circuit itself for reconstruction; (3) embedding regularization via KL against a standard Gaussian prior ($\mathbf{Z}$-Reg.); and (4) joint data-embedding log-likelihood regularization ($p(\mathbf{x}, \mathbf{z})$). For each configuration, Table 1 reports MSE and downstream classification accuracy (DS-Acc.) under full evidence (Full Evi.) conditions, along with the mean MCAR-style corruptions (MCAR) for both MNIST and CIFAR datasets. We deliberately selected these two datasets to evaluate our components across different complexity levels: MNIST serves as a relatively simple benchmark with clear digit structures, while CIFAR represents a more challenging natural image dataset with complex objects, backgrounds, and color variations. We maintain consistent model capacity throughout the experiments to ensure fair comparison across all configurations. Specifically for configurations where NN-Dec. = ✗ (indicating the absence of a neural decoder), we double the encoder size to match the total parameter count of a complete APC model, where capacity is evenly distributed between encoder and decoder.

The ablation results in Table 1 highlight the indispensable role of each component in the APC framework. The baseline configuration (first row) is a circuit that performs both encoding and decoding while being optimized solely through reconstruction error minimization, mimicking a vanilla autoencoding scheme without additional objectives. Differentiable sampling (Diff.) is fundamental, as it ensures end-to-end gradient flow from the neural decoder back to the tractable encoder, optimizing the learned representations for reconstruction.

We evaluate differentiable sampling with SIMPLE "✓" and with Gumbel-Softmax "(✓)". Without this connectivity, training fails to propagate meaningful gradients, resulting in catastrophic degradation: MSE increases by factors of $13.4\times$ on MNIST and $11.9\times$ on CIFAR. Similarly, the neural decoder (NN-Dec.) plays a pivotal role in reconstruction and downstream performance, as its absence leads to severe drops in reconstruction quality, increasing MSE by $5\times$ and $3.7\times$ on MNIST and CIFAR, respectively. This impact is even more pronounced in corruption MCAR metrics, demonstrating the decoder's importance for robustness against missing data. Embedding regularization (**Z**-Reg.) further refines representation learning, nearly halving the MSE on MNIST (from 8.18 to 4.13) and significantly improving the corruption MCAR MSE from 8.43 to 5.06. These results confirm that each component makes a distinct and complementary contribution to the APC framework, with their combination yielding better performance than any partial implementation.

## 6    Conclusion

In this work, we introduced autoencoding probabilistic circuits, a novel framework for representation learning in PCs that leverages their tractability to model explicit probabilistic embeddings. Autoencoding approaches such as VAEs typically rely on neural networks, which often yield implicit and intractable probabilistic formulations. In contrast, APCs model the joint distribution of data and embeddings using a PC encoder. This enables tractable conditional sampling for encoding, yielding embeddings that are samples from a well-defined conditional probability distribution. Our hybrid architecture combines a PC encoder with a neural network decoder, leveraging the complementary strengths of PCs for tractable probabilistic encoding and the modeling capacity of neural networks for decoding. Our empirical evaluations across a wide range of image and tabular datasets demonstrate the effectiveness of APCs. Crucially, APCs show superior reconstruction performance compared to existing circuit-based autoencoding schemes, and to neural encoders in the face of increasing data corruption and missing values. Furthermore, embeddings learned by APCs prove to be robust and meaningful, maintaining downstream task performance even under high data corruption.

**Future Work.**    While our study has thoroughly examined the APC framework, numerous avenues remain for further research. Using a circuit-based encoder grants explicit control over both data and embedding distributions, lifting the typical restriction to independent Gaussians, as seen in VAEs. Moreover, the circuit structures employed in this work were selected heuristically. However, the broader literature on PCs offers various approaches for learning circuit structures. Incorporating these methods could enable more strategic placement of embedding input units, potentially distributing them across different circuit levels. This could facilitate the formation of low-level and high-level representations, introducing hierarchical structures in the learned embedding space. Furthermore, Section 5.4 shows that while modern VAEs achieve high-quality reconstructions on complete data using novel scaling and training tricks, they remain vulnerable to corruption. We could thus apply analogous methodologies to scale PC encoders, aiming for NVAE-level reconstruction fidelity on complete data while preserving APCs' inherent robustness to data corruptions.

### Acknowledgments

This work has benefited from the DYNAMIC center, which is funded by the LOEWE program of the Hessian Ministry of Science and Arts (Grant Number: LOEWE1/16/519/03/09.001(0009)/98), as well as the early stages of the cluster projects by the Deutsche Forschungsgemeinschaft (DFG, German Research Foundation) under Germany's Excellence Strategy—"Reasonable AI" (EXC-3057) and "The Adaptive Mind" (EXC-3066); funding will begin in 2026. Antonio Vergari was funded by the "UNREAL: Unified Reasoning Layer for Trustworthy ML" project (EP/Y023838/1) selected by the ERC and funded by UKRI EPSRC.

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

# A   Encoding Algorithm

---

**Algorithm 1** APC Encoding Procedure

---

**Require:** Probabilistic circuit $\mathcal{C}$ over data $\mathbf{X}$ and embeddings $\mathbf{Z}$, an input data point $\mathbf{x} \in \mathbf{X}$

1:
2: **procedure** ENCODE($\mathcal{C}$: circuit, $\mathbf{x}$: data)
3:     FORWARD(ROOT($\mathcal{C}$), $\mathbf{x}$)                                     ▷ Forward pass; at each sum unit we store its inputs
4:     $[\mathbf{x}, \mathbf{z}]$ = SAMPLE(ROOT($\mathcal{C}$))                          ▷ Sample from conditioned PC $\mathcal{C}|_{\mathbf{X}=\mathbf{x}}$
5:     **return** $\mathbf{z}$

6:
7: **procedure** FORWARD($n$: unit, $\mathbf{x}$: data)                           ▷ General forward pass for circuit units
8:     **if** $n == \bigwedge$ **then**
9:         **if** $\mathbf{x}_n$ is missing **then**
10:             **return** $1.0$                                            ▷ Marginalize missing inputs
11:         **else**
12:             **return** $p_n(\mathbf{x}_n)$                                    ▷ Evaluate PDF of input units
13:     $\boldsymbol{\gamma} \leftarrow \big[\text{FORWARD}(c, \mathbf{x}_c) \text{ for } c \in \text{in}(n)\big]$        ▷ Collect inputs for sum/product units
14:     $n.\boldsymbol{\gamma} \leftarrow \boldsymbol{\gamma}$                                    ▷ Cache inputs for conditional sampling pass
15:     **if** $n == \bigotimes$ **then**
16:         **return** $\prod_{c \in \text{in}(n)} \boldsymbol{\gamma}_c$                              ▷ Product unit
17:     **if** $n == \bigoplus$ **then**
18:         **return** $\sum_{c \in \text{in}(n)} \boldsymbol{\gamma}_c \cdot \theta_n^c$                        ▷ Sum unit

19:
20: **procedure** SAMPLE($n$: unit)                                   ▷ Sampling pass for circuit units
21:     **if** $n == \bigwedge$ **then**
22:         **return** $\mathbf{x}_n \sim p_n(\mathbf{X}_n)$                         ▷ Sample from PDF represented by input unit
23:     **if** $n == \bigotimes$ **then**
24:         **return** CONCAT($\big[\text{SAMPLE}(c) \text{ for } c \in \text{in}(n)\big]$)           ▷ Product unit simply sample all inputs
25:     **if** $n == \bigoplus$ **then**
26:         $\boldsymbol{\theta}_n' \leftarrow$ CONDITION_WEIGHTS($\boldsymbol{\theta}_n, n.\boldsymbol{\gamma}$)        ▷ Condition weights on forward pass likelihoods
27:         $\mathbf{s} \leftarrow$ SIMPLE($\boldsymbol{\theta}_n'$)                              ▷ Sample one-hot encoded index
28:         **return** DOT($\mathbf{s}, \big[\text{SAMPLE}(c) \text{ for } c \in \text{in}(n)\big]$)         ▷ Dot product indexes input samples

29:
30: **procedure** CONDITION_WEIGHTS($\boldsymbol{\theta}$: weights, $\boldsymbol{\gamma}$: inputs)
31:     **for** $i \in 1 \ldots \text{SIZE}(\boldsymbol{\theta})$ **do**
32:         $\theta_i \leftarrow \theta_c \cdot \gamma_i$                                    ▷ Reweight weights based on input likelihoods
33:     $s = \sum_i \theta_i$                                            ▷ Compute normalization constant
34:     **for** $i \in 1 \ldots \text{SIZE}(\boldsymbol{\theta})$ **do**
35:         $\theta_i \leftarrow \theta_i / s$                                    ▷ Normalize weight distribution
36:     **return** $\boldsymbol{\theta}$

---

Conditional sampling in PCs, which forms the basis of our encoding procedure (Appendix Algorithm 1), consists of two passes through the circuit. First, a forward pass computes the marginal likelihood of the evidence, $p_{\mathcal{C}}(\mathbf{x})$, for a given data sample $\mathbf{x}$. During this pass, likelihoods $\boldsymbol{\gamma}$ are cached at the inputs of each sum unit. Subsequently, a sampling pass traverses the circuit from the root to the input units. At each sum unit, it uses the cached likelihoods $\boldsymbol{\gamma}$ from the forward pass to reweight the mixture parameters, effectively applying Bayes' rule to form a posterior over its inputs. It then samples a path according to these conditioned weights using the differentiable SIMPLE estimator. This process yields a sample from the conditional distribution $p_{\mathcal{C}}(\mathbf{Z} \mid \mathbf{X} = \mathbf{x})$, which serves as the embedding $\mathbf{z}$.

# B   Experiment and Evaluation Protocol: Additional Details

All experiments and models are implemented in PyTorch (Ansel et al., 2024) and PyTorch Lightning (Falcon & The PyTorch Lightning team, 2019). Our implementation is available as open-source software at `https://github.com/ml-research/autoencoding-probabilistic-circuits`.

**Datasets**   We evaluate our models on various image and tabular datasets. For image data, we include MNIST (LeCun et al., 1998), Fashion MNIST (F-MNIST) (Xiao et al., 2017), CIFAR-10 (Krizhevsky, 2009), CelebA (Liu et al., 2015), SVHN (Netzer et al., 2011), Flowers (Gurnani et al., 2017), LSUN (Church) (Yu et al., 2015), and Tiny-ImageNet (Deng et al., 2009). Note that Tiny-ImageNet is occasionally abbreviated as ImageNet in our tables for brevity. For tabular data, we utilize the 20 datasets from the binary density estimation benchmark DEBD (Lowd & Davis, 2010; Haaren & Davis, 2012; Bekker et al., 2015; Larochelle & Murray, 2011).

**Models**   In our empirical evaluation, we compare the proposed APCs against several models to demonstrate their effectiveness. These include SPAE-ACT and SPAE-CAT (Vergari et al., 2018) as a circuit-based comparison, as well as vanilla variational autoencoder (VAE) and, to also include more missing-data specific methods MIWAE (Mattei & Frellsen, 2019) and missForest (Stekhoven & Bühlmann, 2011). For all autoencoding models, we use an embedding dimension $d$ of 64 for MNIST and F-MNIST, 256 for all other image-based datasets, and 4,8,16,32, and 64 for tabular datasets, depending on their number of features to ensure $d \ll |\mathbf{X}|$. All models are trained under the same conditions with common and well-established practices.

**Metrics**   We report MSE ($\downarrow$) to measure reconstruction fidelity and SSIM ($\uparrow$) to assess perceptual quality based on structural and visual similarity. To assess encoding robustness, we analyze MSE and SSIM across varying levels of MCAR-style input data corruption, where uniformly random corruption is incrementally increased from 0% to 95% in steps of 5%. We then compute the average reconstruction error over all corruption levels for each metric. To ensure that results are not specific to MCAR-style missing data, Appendix Table 4 presents evaluations using nine additional MAR-style corruption variants.

**Missing Data**   For vanilla VAEs, missing values are imputed using a constant zero value. We also explored alternative imputation strategies, including mean imputation and learned normalization (similar to LayerNorm). In our experiments on MNIST, mean imputation offered improved reconstruction quality at the cost of a reduction in downstream task accuracy. For more challenging datasets like SVHN and CIFAR, the choice of imputation method did not yield statistically significant differences in performance. All other methods handle missing data without the need for imputation.

**Model Capacity**   For all experiments, we ensure that the model encoder and decoder networks are approximately the same size and depth across all models. We use the same neural decoder architecture across APCs and VAE models to ensure that performance differences can be solely attributed to the encoder.

**Circuit Input Units**   For circuit-based encoders, we model data distributions on images with Binomial input units, and Bernoulli input units for the DEBD tabular datasets. Embedding distributions are chosen as Gaussian input units in all cases. We want to highlight once more that the APC framework, in principle, allows for the choice of arbitrary data and *embedding* distributions.

**Sum-Product Autoencoding**   For SPAE, we retrieve the correctly sized embedding vector from a layer in the circuit graph, which consists of exactly $d$ units (embedding size). In addition, when evaluating reconstructions, we report results only for SPAE (instead of SPAE-ACT and SPAE-CAT) because the reconstruction output is identical for both SPAE variants, regardless of whether activation embeddings (ACT) or categorical embeddings (CAT) are used. This equivalence is formally established in Proposition 3 of Vergari et al. (2018). However, in all other evaluations, we distinguish between SPAE-ACT and SPAE-CAT, since their embeddings are inherently different when used in e.g., downstream tasks.

**Autoencoder Training**   Each model is trained for 10,000 iterations using the AdamW optimizer (Kingma & Ba, 2015; Loshchilov & Hutter, 2017), and convergence was confirmed for all models by the end of this training period. We use the MSE as $\mathcal{L}_{\text{REC}}$ for all models. Training is carried out with a batch size of 512, except for CelebA where a batch size of 256 is used due to larger model sizes and VRAM constraints. The initial learning rate is set to 0.1 for APCs and 0.005 for AEs and VAEs. The learning rate is reduced by a factor of 10 at 66% and 90% of the training progress. Additionally, to enhance stability and avoid numerical issues or exploding gradients during the training phase, we utilize an exponential learning rate warmup over the first 2% of training iterations. Empirically, this approach mitigates random training difficulties without affecting the final performance across different random seeds. While, for APCs, we could potentially pretrain the PC-encoder with MLE on the marginal $p(\mathbf{X})$ to speed up convergence, we refrain from doing so in our experiments, to keep the comparison between APC and VAE-based models as fair as possible. All models, with the exception of MIWAE, were trained on the complete datasets without missing values. For MIWAE, training was performed using MCAR-style data in which 50% of entries were missing.

**Downstream Task Training**   We train the logistic regression downstream task models with a batch size of 512 for 5,000 iterations on dataset embeddings from the respective encoders. We employed an initial learning rate of 0.05, which was reduced by a factor of 0.1 at 66% and 90% of the training progress, along with an exponential learning rate warmup over the first 2% of iterations. For SPAE-ACT models specifically, we found it necessary to use the AdamW optimizer with a learning rate of 0.01 for 100,0000 iterations ($20\times$ longer) and normalize the embeddings to achieve comparable downstream task performance better than a random guessing baseline.

**Visualization and Sampling Protocol.**   When generating samples for the baseline PC models (e.g., in Fig. 9), we follow the protocol described in the recent addendum to Peharz et al. (2020a) on arXiv in their Appendix Section D. Specifically, we utilize the standard top-down sampling procedure to hierarchically select the active embedding leaves in the circuit. However, to produce the final image, we use the means of the decoder. As noted by Peharz et al. (2020a), this procedure is directly analogous to the common practice in VAEs where decoder noise is turned off during visualization. In both models, under the assumption of pixel-wise independence in the output distributions, sampling adds independent noise that degrades visual quality without reflecting the model's structural generative capabilities.

## B.1   Model Architectures

**Neural Encoder and Decoder.**   For tabular data, we employ a simple linear feed-forward style network with four hidden layers and leaky ReLU ($\alpha = 0.1$) activations as encoder and decoder. We use convolutional and residual layers with ReLU activations instead for image data. For further details, we refer the reader to `apc.models.{encoder,decoder}.nn_{encoder,decoder}.py` files in our source code repository.

**EinsumNetwork Encoder for Tabular Data.**   We use EinsumNetworks (Peharz et al., 2020a) as circuit structure for APCs and SPAEs with Bernoulli input units for the tabular DEBD dataset (Lowd & Davis, 2010; Haaren & Davis, 2012; Bekker et al., 2015; Larochelle & Murray, 2011). For APCs, embedding input units are inserted at the lowest layer and randomly shuffled with the data distribution input units. For all experiments, we keep a depth of 4, with a single repetition and 32 input unit distributions, and the sum units per scope at each layer.

**ConvPc Encoder for Image Data.**   For image data, we construct a simple "convolutional" circuit for APCs and SPAEs, where we first map each pixel to its density represented by an input unit layer with 256 output channels (number of input unit distributions per scope). We then successively reduce the height and width by a factor of 2 with product layers, which builds the product over all scopes in disjoint neighboring windows (think of non-overlapping convolution windows with stride being the window size). After each product layer, a sum layer maps all channels of each scope to a vector of sum unit outputs (think of convolution in- and out-channels). We repeat this until we end up with a single dimension where the scopes cover the full image input. This allows us to choose a certain depth (how often do we want to repeat the product-sum combination), and the choice of number of sum output units per sum layer. Embedding input units are

inserted randomly at the lowest layer with product units attached to $|\mathbf{Z}|$ data input units, constructing $|\mathbf{Z}|$ combinations of $p_{\mathcal{C}}(x_i)\,p_{\mathcal{C}}(z_j)$. More advanced strategies for embedding insertions remain an avenue for future investigation; for example, embedding input units could be decoupled and integrated at varying hierarchical levels within the circuit.

## C  Reconstruction Performance: Additional Results

This section provides a more detailed exposition of the reconstruction capabilities of APCs and the compared models, supplementing the empirical evaluations presented in the main body. We include additional quantitative results, such as SSIM for image datasets, further explore the impact of various MAR corruption patterns beyond the MCAR scenarios discussed earlier, and present extended qualitative results through more comprehensive sets of reconstruction visualizations across all datasets and corruption types. Furthermore, we offer an initial exploration into the potential of APC embeddings for out-of-distribution detection by analyzing embedding likelihoods.

### C.1  Reconstructions: Comparing Circuit and Neural Encoder (Quantitative Results)

In Section 5.1, we visually demonstrated the robustness of APCs against increasing data corruption by plotting reconstruction error against the percentage of missing values (cf. Fig. 5). To supplement this visual analysis with a direct numerical comparison, we additionally provide detailed quantitative results. Appendix Table 2 presents the average MSE across all MCAR corruption levels for the image datasets, numerically aggregating and substantiating the trends observed in the plots. We further extend this quantitative analysis to tabular data to demonstrate the generalizability of our findings. Appendix Table 3 provides the corresponding reconstruction performance on the 20 DEBD datasets.

Table 2: **APCs achieve lower reconstruction errors than PC, VAE, and missForest baselines across varying levels of MCAR data**. Average reconstruction performance on eight image datasets under increasing percentages of MCAR-style randomly missing pixels (0%-95%).

| | APC | SPAE | VAE | MIWAE | missForest |
|---|---|---|---|---|---|
| MNIST (LeCun et al., 1998) | **5.07** $\pm 0.04$ | 28.46 $\pm 0.43$ | 35.38 $\pm 1.74$ | 9.16 $\pm 0.92$ | 24.51 $\pm 0.63$ |
| F-MNIST (Xiao et al., 2017) | **4.33** $\pm 0.02$ | 18.96 $\pm 0.11$ | 39.80 $\pm 1.10$ | 12.54 $\pm 0.74$ | 24.53 $\pm 0.02$ |
| CIFAR (Krizhevsky, 2009) | **20.65** $\pm 0.09$ | 78.32 $\pm 3.09$ | 55.81 $\pm 0.47$ | 32.80 $\pm 0.27$ | 57.61 $\pm 0.16$ |
| CelebA (Liu et al., 2015) | **166.89** $\pm 0.65$ | 1318.73 $\pm 6.37$ | 1095.90 $\pm 14.0$ | 576.47 $\pm 24.8$ | 1855.67 $\pm 2.29$ |
| SVHN (Netzer et al., 2011) | **4.87** $\pm 0.00$ | 35.38 $\pm 1.92$ | 41.01 $\pm 0.15$ | 17.28 $\pm 0.29$ | 44.19 $\pm 0.25$ |
| Flowers (Gurnani et al., 2017) | **45.22** $\pm 0.14$ | 119.31 $\pm 5.23$ | 107.75 $\pm 11.8$ | 58.03 $\pm 0.37$ | 52.65 $\pm 0.14$ |
| LSUN (Yu et al., 2015) | **63.70** $\pm 0.20$ | 322.22 $\pm 3.48$ | 254.75 $\pm 2.39$ | 123.58 $\pm 1.40$ | 115.28 $\pm 0.15$ |
| ImageNet (Deng et al., 2009) | **118.70** $\pm 0.16$ | 365.06 $\pm 2.20$ | 1286.51 $\pm 10.4$ | 204.10 $\pm 2.04$ | 295.40 $\pm 0.34$ |

Table 3: **APCs achieve lowest reconstruction error on 18 out of 20 tabular dataset.** APCs outperform the PC, VAE and missForest baselines on the 20 DEBD binary tabular dataset and achieve the best reconstruction performance on 18 of the 20 datasets under increasing percentages of MCAR-style randomly missing data (0%-95%).

| | APC | SPAE | VAE | MIWAE | missForest |
|---|---|---|---|---|---|
| accidents | **6.16** $_{\pm 0.04}$ | 12.55 $_{\pm 0.86}$ | 7.56 $_{\pm 0.28}$ | 7.34 $_{\pm 0.05}$ | 6.75 $_{\pm 0.01}$ |
| ad | 5.57 $_{\pm 0.03}$ | 231.84 $_{\pm 15.4}$ | 6.04 $_{\pm 0.17}$ | **3.50** $_{\pm 0.06}$ | 5.57 $_{\pm 0.03}$ |
| baudio | **6.50** $_{\pm 0.03}$ | 15.60 $_{\pm 1.33}$ | 7.44 $_{\pm 0.01}$ | 7.81 $_{\pm 0.04}$ | 7.50 $_{\pm 0.01}$ |
| bbc | **34.66** $_{\pm 0.15}$ | 161.59 $_{\pm 10.8}$ | 37.45 $_{\pm 0.55}$ | 47.08 $_{\pm 0.60}$ | 35.02 $_{\pm 0.15}$ |
| bnetflix | **10.08** $_{\pm 0.02}$ | 20.53 $_{\pm 0.96}$ | 13.49 $_{\pm 0.77}$ | 14.81 $_{\pm 0.04}$ | 10.80 $_{\pm 0.00}$ |
| book | **3.65** $_{\pm 0.02}$ | 67.10 $_{\pm 9.40}$ | 3.73 $_{\pm 0.02}$ | 4.40 $_{\pm 0.06}$ | 3.87 $_{\pm 0.03}$ |
| c20ng | **19.12** $_{\pm 0.04}$ | 123.50 $_{\pm 5.85}$ | 20.69 $_{\pm 0.05}$ | 25.50 $_{\pm 0.28}$ | 20.34 $_{\pm 0.02}$ |
| cr52 | **11.94** $_{\pm 0.10}$ | 134.30 $_{\pm 7.81}$ | 12.83 $_{\pm 0.18}$ | 15.10 $_{\pm 0.19}$ | 12.75 $_{\pm 0.09}$ |
| cwebkb | **21.32** $_{\pm 0.14}$ | 109.07 $_{\pm 4.38}$ | 22.94 $_{\pm 0.15}$ | 28.53 $_{\pm 0.32}$ | 22.71 $_{\pm 0.07}$ |
| dna | **15.89** $_{\pm 0.12}$ | 28.76 $_{\pm 0.91}$ | 16.46 $_{\pm 0.17}$ | 20.46 $_{\pm 0.30}$ | **15.89** $_{\pm 0.04}$ |
| jester | **9.04** $_{\pm 0.02}$ | 20.98 $_{\pm 0.33}$ | 17.44 $_{\pm 0.10}$ | 15.04 $_{\pm 0.08}$ | 10.66 $_{\pm 0.01}$ |
| kdd | **0.19** $_{\pm 0.00}$ | 4.12 $_{\pm 0.57}$ | 0.19 $_{\pm 0.00}$ | 0.22 $_{\pm 0.00}$ | 0.20 $_{\pm 0.00}$ |
| kosarek | **1.35** $_{\pm 0.04}$ | 13.57 $_{\pm 3.63}$ | 1.42 $_{\pm 0.01}$ | 1.67 $_{\pm 0.01}$ | 1.40 $_{\pm 0.00}$ |
| moviereview | **48.06** $_{\pm 0.15}$ | 141.19 $_{\pm 4.27}$ | 50.08 $_{\pm 0.22}$ | 70.07 $_{\pm 0.13}$ | 48.36 $_{\pm 0.06}$ |
| msnbc | 1.05 $_{\pm 0.03}$ | 2.23 $_{\pm 0.39}$ | 1.03 $_{\pm 0.01}$ | 1.17 $_{\pm 0.00}$ | **0.99** $_{\pm 0.00}$ |
| nltcs | **1.12** $_{\pm 0.02}$ | 2.48 $_{\pm 0.00}$ | 1.60 $_{\pm 0.01}$ | 1.37 $_{\pm 0.01}$ | 1.48 $_{\pm 0.00}$ |
| plants | **2.52** $_{\pm 0.04}$ | 11.62 $_{\pm 0.54}$ | 3.93 $_{\pm 0.05}$ | 2.93 $_{\pm 0.01}$ | 4.67 $_{\pm 0.01}$ |
| pumsb_star | **5.82** $_{\pm 0.16}$ | 23.30 $_{\pm 0.74}$ | 12.47 $_{\pm 0.40}$ | 6.53 $_{\pm 0.10}$ | 11.74 $_{\pm 0.02}$ |
| tmovie | **7.32** $_{\pm 0.06}$ | 57.94 $_{\pm 5.49}$ | 8.72 $_{\pm 0.03}$ | 9.04 $_{\pm 0.04}$ | 10.11 $_{\pm 0.02}$ |
| tretail | **1.22** $_{\pm 0.01}$ | 6.22 $_{\pm 1.63}$ | 1.23 $_{\pm 0.00}$ | 1.58 $_{\pm 0.02}$ | **1.22** $_{\pm 0.00}$ |
| voting | **27.71** $_{\pm 0.18}$ | 264.53 $_{\pm 7.30}$ | 122.24 $_{\pm 52.8}$ | 39.04 $_{\pm 0.46}$ | 119.03 $_{\pm 0.12}$ |

## C.2 Missing at Random (MAR) Style Corruptions

Table 4: **APCs ranks first on all MAR-style corruption types across all datasets.** Model ranking per corruption type, aggregated across datasets. Each row represents a different MAR-style corruption applied to the input images. The severity of each corruption is linearly increased, and the average reconstruction MSE/SSIM is measured. The models are ranked based on their robustness to these corruptions aggregated across all image datasets. The reported values represent the ranking based on the average MSE/SSIM. See Appendix Figures 15 and 16 for reconstruction visualizations.

| | MSE | | | | SSIM | | | |
|---|---|---|---|---|---|---|---|---|
| Corruption | APC | SPAE | VAE | MIWAE | APC | SPAE | VAE | MIWAE |
| left-to-right | **1** | 4 | 2 | 3 | **1** | 3 | 2 | 4 |
| right-to-left | **1** | 4 | 2 | 3 | **1** | 2 | 3 | 4 |
| top-to-bottom | **1** | 4 | 2 | 3 | **1** | 3 | 2 | 4 |
| bottom-to-top | **1** | 4 | 2 | 3 | **1** | 3 | 2 | 4 |
| border-to-center | **1** | 3 | 2 | 4 | **1** | 2 | 3 | 4 |
| center-to-border | **1** | 3 | 4 | 2 | **1** | 2 | 3 | 4 |
| horizontal-band | **1** | 3 | 4 | 2 | **1** | 4 | 2 | 3 |
| vertical-band | **1** | 2 | 3 | 4 | **1** | 3 | 2 | 4 |
| salt-and-pepper | **1** | 3 | 2 | 4 | **1** | 4 | 2 | 3 |

Table 5: **APCs achieve higher SSIM scores compared to PC, VAE, and missForest baselines across varying levels of MCAR data**. Average reconstruction SSIM performance on eight image datasets under increasing percentages of MCAR-style randomly missing pixels (0%-95%).

| | APC | SPAE | VAE | MIWAE | missForest |
|---|---|---|---|---|---|
| MNIST | **93.33** $\pm0.04$ | 69.87 $\pm0.36$ | 50.74 $\pm0.82$ | 86.81 $\pm1.14$ | 62.29 $\pm0.46$ |
| F-MNIST | **87.40** $\pm0.05$ | 68.89 $\pm0.23$ | 51.75 $\pm0.38$ | 78.24 $\pm0.43$ | 54.00 $\pm0.04$ |
| CIFAR | **78.20** $\pm0.07$ | 50.61 $\pm0.88$ | 61.64 $\pm0.14$ | 74.25 $\pm0.15$ | 57.98 $\pm0.13$ |
| CelebA | **79.52** $\pm0.04$ | 49.02 $\pm0.17$ | 50.30 $\pm0.30$ | 61.26 $\pm0.66$ | 37.16 $\pm0.10$ |
| SVHN | **90.29** $\pm0.03$ | 54.88 $\pm0.91$ | 61.79 $\pm0.18$ | 76.23 $\pm0.47$ | 56.86 $\pm0.17$ |
| Flowers | **60.40** $\pm0.16$ | 44.01 $\pm0.82$ | 50.63 $\pm0.43$ | 56.68 $\pm0.11$ | 56.94 $\pm0.10$ |
| LSUN | **77.14** $\pm0.07$ | 50.73 $\pm0.52$ | 55.63 $\pm0.28$ | 59.80 $\pm0.16$ | 27.32 $\pm0.02$ |
| ImageNet | **71.55** $\pm0.04$ | 50.62 $\pm0.36$ | 47.76 $\pm0.23$ | 61.58 $\pm0.15$ | 50.98 $\pm0.05$ |

To further investigate model robustness beyond *randomly* missing data, Appendix Table 4 presents model rankings across different MAR-style corruption types aggregated over all datasets. For each corruption type, we repeat the experiments presented in Appendix Table 2 for all image datasets, assign each method a rank from best (1.) to worst (4.) and then report their final rank, sorted by the average rank over all datasets. We only compare autoencoding methods, as they reconstruct the full image from a low-dimensional embedding space. In contrast, missForest operates directly in the input space, making it an inappropriate comparison for a ranking in this context. APCs consistently achieve the top rank for all but one corruption type and metric, indicating that their superior performance extends beyond handling missing data to various other forms of data corruption. Vanilla VAEs appear to outrank its missing-data-specific variant MIWAE in both MSE and SSIM measurements. This could be attributed to the fact that MIWAE specifically trains on a specific type of missing data, MCAR in our case. This leads to a decreased performance in reconstruction when the pattern of missing data substantially changes. We can visually confirm this is Appendix Figures 14 and 16, where MIWAE is better at reconstructing MCAR-style missing data than VAEs, but fails similarly when confronted with MAR-style corruptions (horizontal, vertical, center, border).

### C.3 Structured Similarity Index (SSIM) Evaluations

While MSE provides a quantitative measure of pixel-wise differences, SSIM is also employed to offer a complementary perspective on reconstruction quality. SSIM is designed to better align with human visual perception by evaluating luminance, contrast, and structure similarities between the original and reconstructed images. Consistent with the MSE results, the findings presented in Appendix Table 5 (complementary to Appendix Table 2) and Appendix Figure 12 (complementary to Fig. 5) demonstrate that APCs significantly outperform all other compared methods in preserving image fidelity across all datasets, even as the proportion of missing data increases.

### C.4 Additional Reconstruction Visualizations

To provide further visual insights and extend the quantitative reconstructions presented in Fig. 6, we additionally show MCAR-style corruptions for 0%, 30%, 50%, 70%, and 90% uniformly random missing data in Appendix Figures 13 and 15 and horizontal bands, vertical band, center and border MAR-style corruptions in Appendix Figures 14 and 16 on all image datasets for all methods. These results further provide evidence that APCs outperform the alternative methods in all investigated corruption scenarios, providing better reconstructions across the bench.

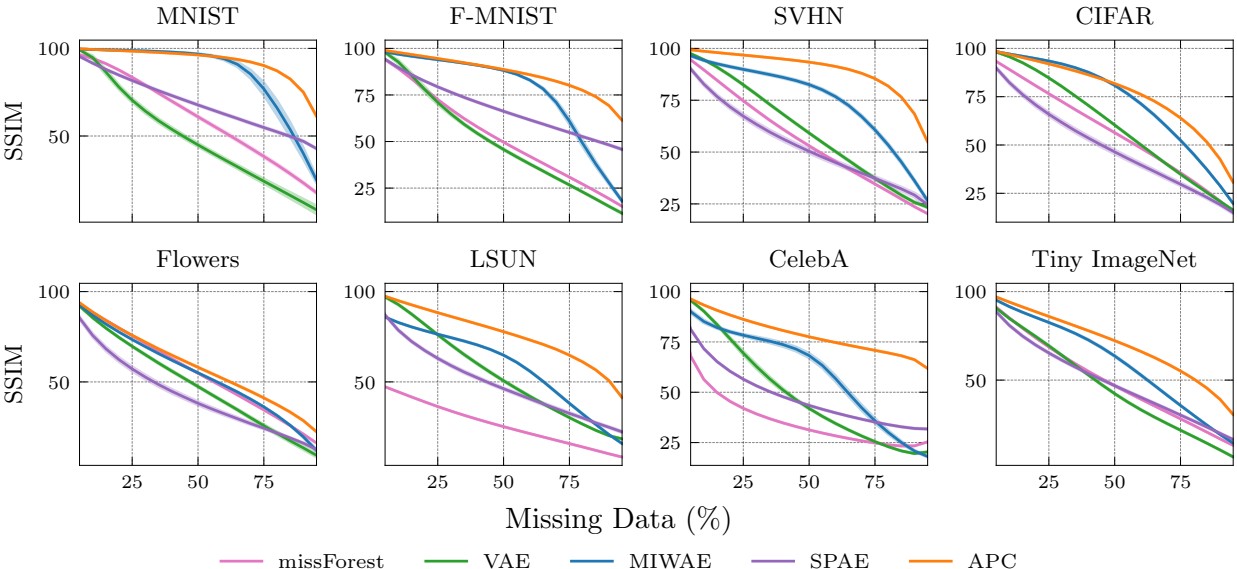

Figure 12: **APCs can deliver higher reconstruction SSIM scores compared to PC, VAE, and missForest baselines in the asymptotic regime of MCAR-style randomly missing data**. Reconstruction SSIM is reported across all image datasets as the degree of MCAR corruption increases from 0% to 95%.

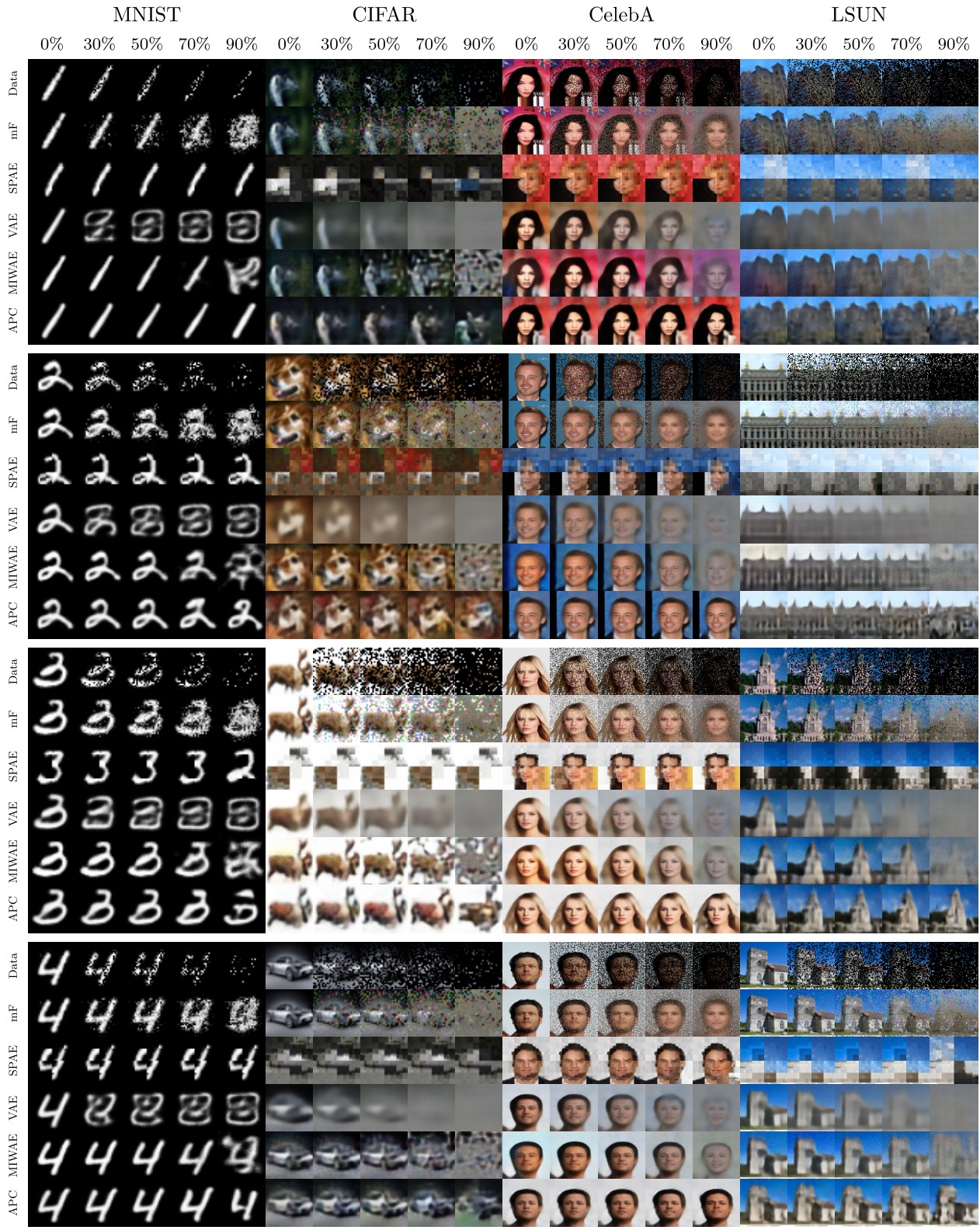

Figure 13: Reconstructions of various MCAR corruptions for MNIST, CIFAR, CelebA, and LSUN.

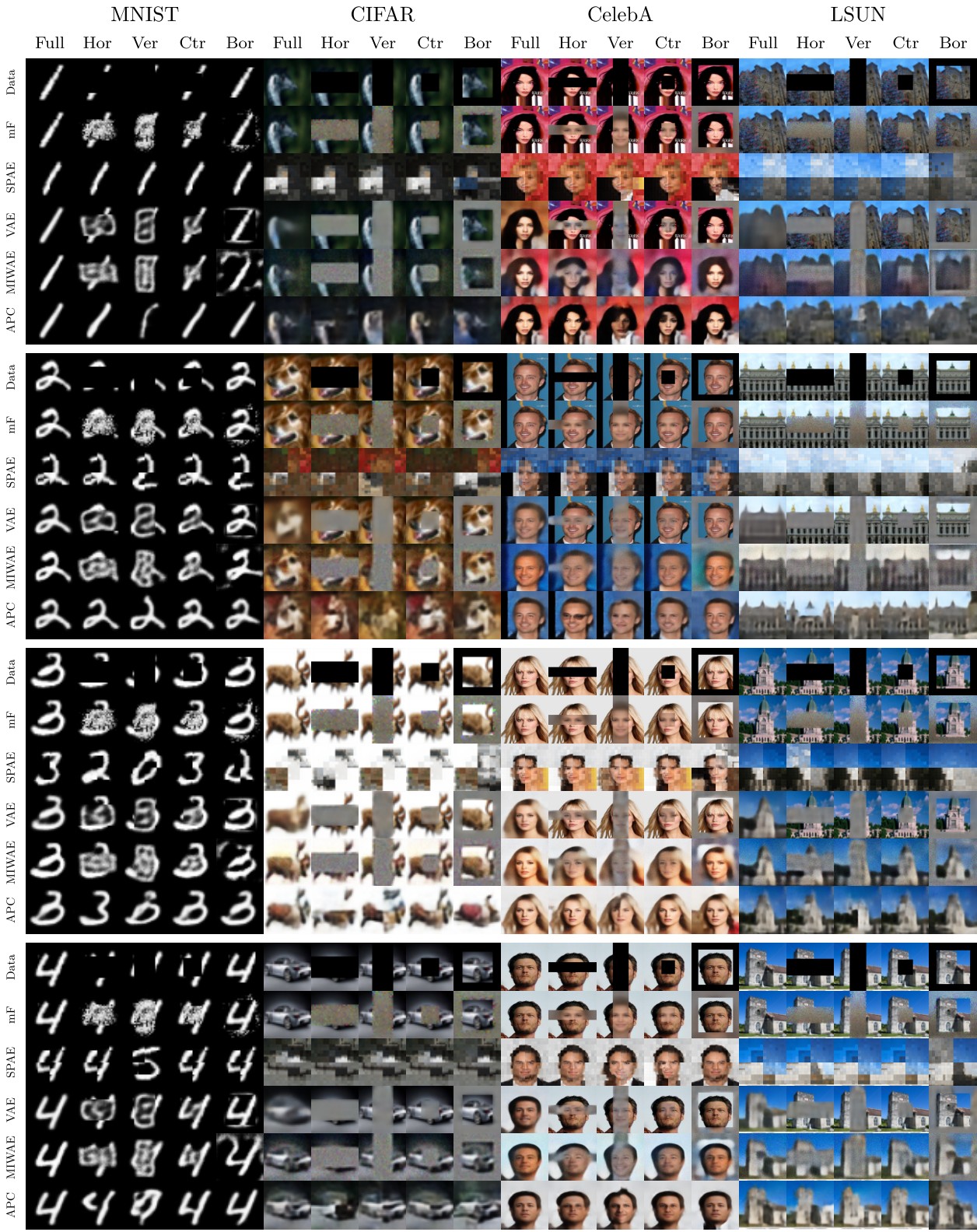

Figure 14: Reconstructions of various MAR corruptions for MNIST, CIFAR, CelebA, and LSUN.

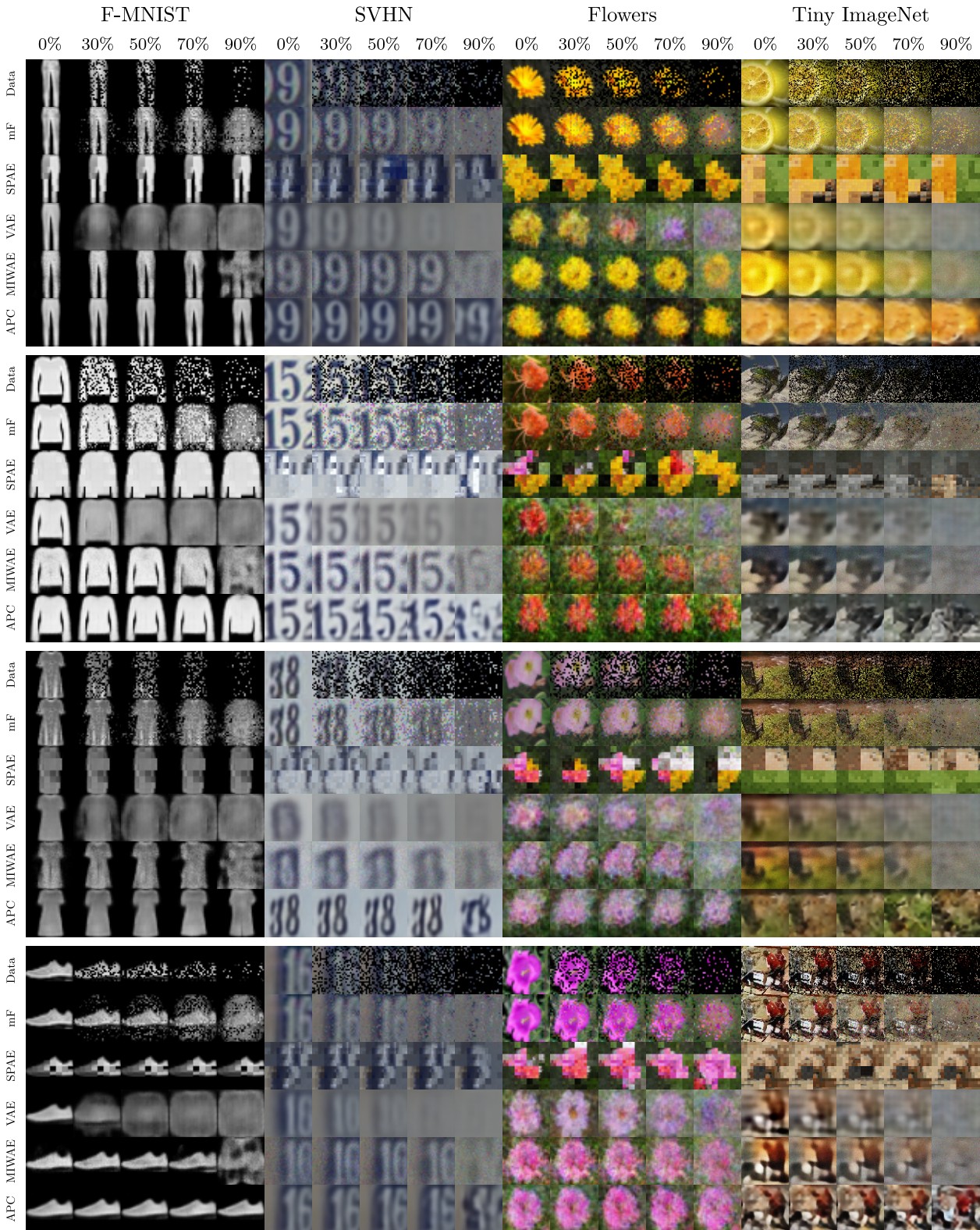

Figure 15: Reconstructions of various MCAR corruptions for F-MNIST, SVHN, Flowers, and Tiny-ImageNet.

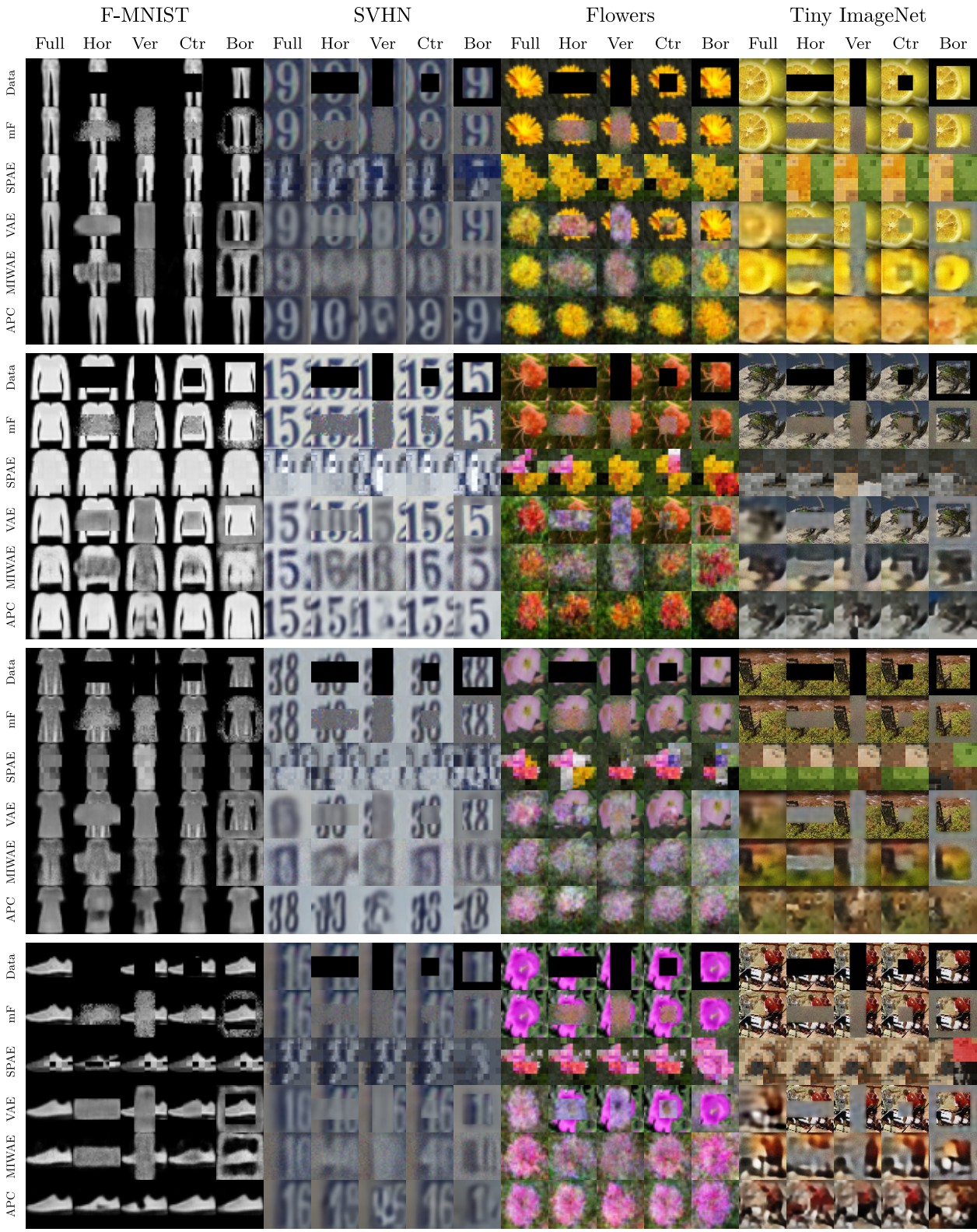

Figure 16: Reconstructions of various MAR corruptions for F-MNIST, SVHN, Flowers, and Tiny-ImageNet.

# D  Autoencoding Probabilistic Circuit Variants

Having established the contribution of each component through our ablation studies in Section 5.6, we now build upon the core APC framework and additionally explore several alternative configurations motivated by previous work of sum-product autoencoding (Vergari et al., 2018). These variations allow us to examine the impact of different encoding and decoding strategies within the APC framework. Specifically, we consider the following:

- **APC $_{\text{pc}}$**: We replace the neural network decoder with the circuit encoder $\mathcal{C}$ itself. Leveraging the flexibility of PCs, reconstructions $\hat{\mathbf{x}}$ are generated by sampling from the conditional distribution $p_{\mathcal{C}}(\mathbf{X} \,|\, \mathbf{Z})$. Differentiable sampling enables end-to-end training of this configuration, using the same objective as standard APCs to optimize the circuit for both encoding and decoding.

- **APC-SPAE$_{\text{pc}}$**: This configuration trains an APC using the SPAE scheme for encoding and decoding, employing the circuit $\mathcal{C}$ for both. This means the encoding and decoding processes are based on ACT embeddings (CAT embeddings lead to identical reconstructions as per Vergari et al. (2018)). Note, that KL regularization of embeddings is omitted due to the different nature of ACT and CAT embeddings compared to the default APC formulation, and $\mathcal{L}_{\text{NLL}}$ is limited to $p_{\mathcal{C}}(\mathbf{X})$, as explicit embedding representations $\mathbf{Z}$ are not directly modeled in the circuit in this variant.

- **APC-SPAE-ACT$_{\text{nn}}$**: We combine the ACT encoding from SPAE with a neural decoder. Similar to APC-SPAE$_{\text{pc}}$, KL regularization is not applied and $\mathcal{L}_{\text{NLL}}$ is limited to $p_{\mathcal{C}}(\mathbf{X})$. In our experiments, we found that due to circuit activation statistics, it was necessary to add a batch normalization layer on top of the circuit activations for the training with a neural decoder to be stable.

- **APC-SPAE-CAT$_{\text{nn}}$**: Analogous to APC-SPAE-ACT$_{\text{nn}}$, this variant utilizes the CAT embedding encoding approach from SPAE, coupled with a neural decoder for reconstruction. Training follows the same strategy as APC-SPAE-ACT$_{\text{nn}}$, excluding KL regularization and limiting $\mathcal{L}_{\text{NLL}}$ to $p_{\mathcal{C}}(\mathbf{X})$.

In Appendix Tables 6 and 7, we report the reconstruction performance of all APC variants on image and tabular datasets with progressively increasing proportions of randomly missing pixels, contrasting them with full APC and SPAE models. We evaluate reconstruction quality using the average MCAR-style corruption reconstruction error for both MSE and SSIM, which capture the impact of varying corruption levels. As shown, APCs maintain the overall best performance across all datasets. However, we make three key observations. (1) Utilizing the encoding PC directly as a decoder (APC $_{\text{pc}}$) yields inferior performance across all tasks, with a median relative decrease of $7.3\times$ on tabular data $2.7\times$ in image data reconstruction, compared to employing a neural network decoder, highlighting, that the additional modeling capacity and flexibility of a neural decoder is crucial. (2) Integrating the SPAE scheme within the APC framework (APC-SPAE$_{\text{pc}}$) also led to a reduction in performance, further suggesting that a PC decoder is less effective in this context. (3) Incorporating SPAE-ACT and SPAE-CAT encoding mechanisms into the APC framework, while retaining a neural network decoder, consistently improved upon the performance of the baseline SPAE method in all evaluated scenarios, achieving median improvements of $2.5\times$ for ACT and $1.7\times$ for CAT embeddings on image data, and median $3.9\times$ and $4.8\times$ respectively on tabular datasets. We use median values here to reduce the impact of extreme outliers, which we obtained on some of the datasets.

We want to highlight that CAT embeddings (Vergari et al., 2018) can be interpreted as a special case of APCs. They are the result of running `MaxProdMPE` on the circuit $\bar{\mathcal{C}}$ over $\mathbf{V} = (\mathbf{X}, \mathbf{H})$ where $\mathbf{H}$ are sum unit indicator random variables obtained from the latent variable interpretation introduced in Peharz et al. (2017) and described in Section 2. In contrast, the APC framework allows embedding random variables to appear at *arbitrary positions* in the circuit graph represented by *arbitrary distributions*. This comparison highlights the flexibility of APCs compared to the prior autoencoding scheme introduced in Vergari et al. (2018).

Table 6: **APCs achieve lower reconstruction errors than PC and SPAE-based variants across varying levels of MCAR data**. Average reconstruction performance on eight image datasets under increasing percentages of MCAR-style randomly missing pixels (0%-95%).

| | APC | APC$_{pc}$ | SPAE | APC-SPAE$_{pc}$ | APC-SPAE-ACT$_{nn}$ | APC-SPAE-CAT$_{nn}$ |
|---|---|---|---|---|---|---|
| | | | MSE $\downarrow$ | | | |
| MNIST | **5.07** $\pm 0.04$ | 17.17 $\pm 0.41$ | 28.46 $\pm 0.43$ | 49.82 $\pm 3.01$ | 14.71 $\pm 0.25$ | 20.07 $\pm 1.00$ |
| F-MNIST | **4.33** $\pm 0.02$ | 17.09 $\pm 0.89$ | 18.96 $\pm 0.11$ | 49.46 $\pm 2.75$ | 12.79 $\pm 0.62$ | 17.97 $\pm 3.58$ |
| CIFAR | **20.65** $\pm 0.09$ | 59.22 $\pm 0.30$ | 78.32 $\pm 3.09$ | 102.74 $\pm 0.93$ | 31.88 $\pm 0.53$ | 42.44 $\pm 0.26$ |
| CelebA | **166.89** $\pm 0.65$ | 644.36 $\pm 14.2$ | 1318.73 $\pm 6.37$ | 2969.30 $\pm 235.$ | 1173.57 $\pm 156.$ | 1220.76 $\pm 318.$ |
| SVHN | **4.87** $\pm 0.00$ | 32.27 $\pm 0.38$ | 35.38 $\pm 1.92$ | 78.01 $\pm 4.37$ | 17.59 $\pm 0.24$ | 18.74 $\pm 0.35$ |
| Flowers | **45.22** $\pm 0.14$ | 58.65 $\pm 0.72$ | 119.31 $\pm 5.23$ | 110.05 $\pm 6.19$ | 72.46 $\pm 0.60$ | 75.05 $\pm 2.91$ |
| LSUN | **63.70** $\pm 0.20$ | 185.92 $\pm 3.62$ | 322.22 $\pm 3.48$ | 489.87 $\pm 18.7$ | 194.10 $\pm 16.6$ | 185.69 $\pm 3.11$ |
| ImageNet | **118.70** $\pm 0.16$ | 276.92 $\pm 3.07$ | 365.06 $\pm 2.20$ | 566.89 $\pm 17.2$ | 369.20 $\pm 35.7$ | 267.96 $\pm 20.6$ |
| | | | SSIM $\uparrow$ | | | |
| MNIST | **93.33** $\pm 0.04$ | 79.87 $\pm 0.57$ | 69.87 $\pm 0.36$ | 45.21 $\pm 3.15$ | 82.63 $\pm 0.35$ | 64.52 $\pm 2.46$ |
| F-MNIST | **87.40** $\pm 0.05$ | 69.44 $\pm 1.00$ | 68.89 $\pm 0.23$ | 39.11 $\pm 1.18$ | 75.81 $\pm 0.58$ | 60.89 $\pm 4.83$ |
| CIFAR | **78.20** $\pm 0.07$ | 58.23 $\pm 0.31$ | 50.61 $\pm 0.88$ | 44.18 $\pm 0.17$ | 71.41 $\pm 0.18$ | 61.31 $\pm 0.14$ |
| CelebA | **79.52** $\pm 0.04$ | 58.79 $\pm 0.33$ | 49.02 $\pm 0.17$ | 26.71 $\pm 1.17$ | 49.03 $\pm 0.83$ | 44.56 $\pm 3.77$ |
| SVHN | **90.29** $\pm 0.03$ | 62.82 $\pm 0.24$ | 54.88 $\pm 0.91$ | 39.22 $\pm 0.76$ | 75.60 $\pm 0.25$ | 67.86 $\pm 0.36$ |
| Flowers | **60.40** $\pm 0.16$ | 59.16 $\pm 0.15$ | 44.01 $\pm 0.82$ | 40.80 $\pm 0.87$ | 50.35 $\pm 0.18$ | 51.86 $\pm 1.12$ |
| LSUN | **77.14** $\pm 0.07$ | 59.09 $\pm 0.38$ | 50.73 $\pm 0.52$ | 37.45 $\pm 0.93$ | 58.72 $\pm 0.89$ | 55.39 $\pm 0.39$ |
| ImageNet | **71.55** $\pm 0.04$ | 55.68 $\pm 0.29$ | 50.62 $\pm 0.36$ | 37.90 $\pm 0.77$ | 55.43 $\pm 0.45$ | 53.50 $\pm 0.97$ |

Table 7: **APCs achieve lower reconstruction errors than PC and SPAE-based variants across varying levels of MCAR data**. Average reconstruction performance on the 20 DEBD binary tabular datasets under increasing percentages of MCAR-style randomly missing pixels (0%-95%).

| | APC | APC$_{pc}$ | SPAE | APC-SPAE$_{pc}$ | APC-SPAE-ACT$_{nn}$ | APC-SPAE-CAT$_{nn}$ |
|---|---|---|---|---|---|---|
| accidents | **6.16** $\pm 0.04$ | 29.16 $\pm 0.52$ | 12.55 $\pm 0.86$ | 22.16 $\pm 1.95$ | 8.31 $\pm 0.31$ | 6.75 $\pm 0.01$ |
| ad | **5.57** $\pm 0.03$ | 346.14 $\pm 5.94$ | 275.00 $\pm 68.1$ | 374.18 $\pm 15.7$ | 7.88 $\pm 0.41$ | **5.57** $\pm 0.03$ |
| baudio | **6.50** $\pm 0.03$ | 23.31 $\pm 1.31$ | 15.60 $\pm 1.33$ | 16.81 $\pm 0.72$ | 7.27 $\pm 0.16$ | 7.50 $\pm 0.01$ |
| bbc | **34.66** $\pm 0.15$ | 238.77 $\pm 3.22$ | 161.59 $\pm 10.8$ | 236.07 $\pm 4.29$ | 42.77 $\pm 0.75$ | 35.02 $\pm 0.15$ |
| bnetflix | **10.08** $\pm 0.02$ | 24.87 $\pm 0.64$ | 20.53 $\pm 0.96$ | 21.89 $\pm 0.34$ | 10.56 $\pm 0.15$ | 10.80 $\pm 0.00$ |
| book | **3.65** $\pm 0.02$ | 121.44 $\pm 2.65$ | 67.10 $\pm 9.40$ | 95.96 $\pm 4.73$ | 3.89 $\pm 0.18$ | 3.87 $\pm 0.03$ |
| c20ng | **19.12** $\pm 0.04$ | 215.58 $\pm 5.08$ | 123.50 $\pm 5.85$ | 187.31 $\pm 5.87$ | 20.60 $\pm 0.26$ | 20.34 $\pm 0.02$ |
| cr52 | **11.94** $\pm 0.10$ | 201.12 $\pm 5.98$ | 134.30 $\pm 7.81$ | 217.42 $\pm 10.8$ | 14.33 $\pm 0.34$ | 12.75 $\pm 0.09$ |
| cwebkb | **21.32** $\pm 0.14$ | 193.05 $\pm 3.82$ | 109.07 $\pm 4.38$ | 189.50 $\pm 7.03$ | 24.40 $\pm 0.47$ | 22.70 $\pm 0.07$ |
| dna | **15.89** $\pm 0.12$ | 46.77 $\pm 1.38$ | 28.76 $\pm 0.91$ | 34.41 $\pm 1.21$ | 21.19 $\pm 0.45$ | **15.89** $\pm 0.04$ |
| jester | **9.04** $\pm 0.02$ | 24.39 $\pm 0.47$ | 20.98 $\pm 0.33$ | 21.08 $\pm 0.83$ | 10.20 $\pm 0.12$ | 10.66 $\pm 0.01$ |
| kdd | **0.19** $\pm 0.00$ | 13.20 $\pm 1.73$ | 4.12 $\pm 0.57$ | 7.55 $\pm 1.30$ | 0.33 $\pm 0.03$ | 0.20 $\pm 0.00$ |
| kosarek | **1.35** $\pm 0.04$ | 43.28 $\pm 2.04$ | 13.57 $\pm 3.63$ | 36.47 $\pm 3.56$ | 2.15 $\pm 0.11$ | 1.40 $\pm 0.00$ |
| moviereview | **48.06** $\pm 0.15$ | 226.20 $\pm 4.90$ | 141.19 $\pm 4.27$ | 234.37 $\pm 10.9$ | 60.11 $\pm 0.99$ | 48.36 $\pm 0.06$ |
| msnbc | 1.05 $\pm 0.03$ | 2.68 $\pm 0.43$ | 2.23 $\pm 0.39$ | 2.12 $\pm 0.34$ | 1.67 $\pm 0.01$ | **0.99** $\pm 0.00$ |
| nltcs | **1.12** $\pm 0.02$ | 3.26 $\pm 0.27$ | 2.48 $\pm 0.00$ | 2.86 $\pm 0.39$ | 1.66 $\pm 0.14$ | 1.48 $\pm 0.00$ |
| plants | **2.52** $\pm 0.04$ | 16.44 $\pm 1.20$ | 11.62 $\pm 0.54$ | 12.04 $\pm 0.40$ | 3.22 $\pm 0.16$ | 4.67 $\pm 0.01$ |
| pumsb_star | **5.82** $\pm 0.16$ | 42.21 $\pm 1.94$ | 23.30 $\pm 0.74$ | 32.76 $\pm 1.53$ | 8.99 $\pm 0.65$ | 11.74 $\pm 0.02$ |
| tmovie | **7.32** $\pm 0.06$ | 116.74 $\pm 1.89$ | 57.94 $\pm 5.49$ | 90.92 $\pm 10.4$ | 10.39 $\pm 0.34$ | 10.11 $\pm 0.02$ |
| tretail | **1.22** $\pm 0.01$ | 30.08 $\pm 2.20$ | 6.22 $\pm 1.63$ | 34.79 $\pm 4.17$ | 2.12 $\pm 0.25$ | **1.22** $\pm 0.00$ |
| voting | **27.71** $\pm 0.18$ | 311.78 $\pm 3.31$ | 264.53 $\pm 7.30$ | 303.02 $\pm 13.6$ | 91.81 $\pm 6.90$ | 119.05 $\pm 0.11$ |

# E  Knowledge Distillation: Algorithm & Additional Results

---

**Algorithm 2** Data-Free Knowledge Distillation from VAE to APC

---

**Require:** Pre-trained Teacher VAE, Student APC, Iterations $T$

 1: **procedure** KNOWLEDGEDISTILLATION(VAE, APC, $T$)

 2:      **for** $t = 1, 2, \ldots, T$ **do**

 3:          $\mathbf{z} \sim p_{\mathcal{C}}(\mathbf{Z})$                                             ▷ Sample embedding from APC prior

 4:          $\hat{\mathbf{x}}_{\text{VAE}} = g_{\text{VAE}}(\mathbf{z})$                               ▷ Generate synthetic data with VAE decoder

 5:          $\mathbf{z}_{\text{VAE}} = f_{\text{VAE}}(\hat{\mathbf{x}}_{\text{VAE}})$                           ▷ Encode synthetic data with VAE encoder

 6:          $\mathbf{z}_{\text{APC}} \sim p_{\mathcal{C}}(\mathbf{Z} \mid \mathbf{x}_{\text{VAE}})$                        ▷ Encode synthetic data with APC encoder

 7:          $\hat{\mathbf{x}}_{\text{APC}} = g_{\text{APC}}(\mathbf{z}_{\text{APC}})$                           ▷ Reconstruct with APC decoder

 8:          $\mathcal{L} = \log p_{\theta}(\mathbf{x}_{\text{VAE}} \mid \mathbf{z}_{\text{APC}}) + \text{KL}(p_{\mathcal{C}}(\mathbf{Z} \mid \hat{\mathbf{x}}_{\text{VAE}}) \,||\, q(\mathbf{z})) - \log p_{\mathcal{C}}(\hat{\mathbf{x}}_{\text{VAE}}, \mathbf{z}_{\text{VAE}})$   ▷ Distillation loss

 9:          Update APC parameters to minimize $\mathcal{L}$                             ▷ Gradient descent

10:      **return** Distilled APC

---

Appendix Algorithm 2 outlines the process of data-free knowledge distillation from a VAE to an APC of similar capacity and the same decoder architecture. The procedure iteratively refines the APC to learn from the VAE without the need for original training data. At each iteration, an embedding $\mathbf{z}$ is first sampled from the APC prior $p_{\mathcal{C}}(\mathbf{Z})$. This embedding is then passed through the VAE decoder $g_{\text{VAE}}(\mathbf{z})$ to generate synthetic data $\hat{\mathbf{x}}_{\text{VAE}}$ that aligns with the APC prior. The synthetic data is subsequently re-encoded using the VAE encoder $\mathbf{z}_{\text{VAE}} = f_{\text{VAE}}(\hat{\mathbf{x}}_{\text{VAE}})$, to obtain an embedding according to the VAE's approximate posterior. In parallel, the APC encodes the same synthetic data using $p_{\mathcal{C}}(\mathbf{Z} \mid \mathbf{x}_{\text{VAE}})$, obtaining an auxiliary embedding $\mathbf{z}_{\text{APC}}$. This embedding is then passed through the APC decoder $g_{\text{APC}}(\mathbf{z}_{\text{APC}})$ to reconstruct the synthetic VAE sample, producing $\hat{\mathbf{x}}_{\text{APC}}$. The distillation loss $\mathcal{L}$ is equal to the main training recipe of APCs and consists of three components: the mean squared error (MSE) between the VAE and APC reconstructions, a Kullback-Leibler divergence (KL) term that encourages the APC 's posterior $p_{\mathcal{C}}(\mathbf{Z} \mid \hat{\mathbf{x}}_{\text{VAE}})$ to align with a standard normal prior $\mathcal{N}(0, 1)$, and a negative log-likelihood (NLL) term maximizing the joint likelihood of VAE sample and embedding $p_{\mathcal{C}}(\hat{\mathbf{x}}_{\text{VAE}}, \mathbf{z}_{\text{VAE}})$. Since all steps are differentiable, we can end-to-end train this procedure from Line 3 to Line 7 and update the APC parameters via gradient descent to minimize $\mathcal{L}$.

To demonstrate that knowledge distillation from VAEs to APCs generalizes beyond the two exemplary datasets considered in Section 5.5, we extend our evaluation to all image and tabular datasets introduced in Section 5. We report results for both teacher and student reconstructions, and downstream task accuracy under full evidence (Full Evi.) and MCAR-style corruptions in Appendix Table 8.

Table 8: **APCs successfully distill the knowledge from a pre-trained VAE in a data-free setting and exceed their teachers robustness against missing data.** Input reconstruction mean squared error (↓) and downstream task accuracy (↑) performance comparison between VAE teacher and distilled APC student models, evaluated with full evidence (Full Evi.) and under varying levels of missing data (MCAR).

**Full Evidence**: APCs are successfully distilling the knowledge from their teacher.

| | Teacher | → | Student |
|---|---|---|---|
| | MSE (↓) | | |
| MNIST | 3.15 ±0.02 | | 6.84 ±0.17 |
| F-MNIST | 5.84 ±0.02 | | 8.16 ±0.10 |
| CIFAR | 21.84 ±0.07 | | 25.34 ±0.30 |
| CelebA | 336.00 ±22.7 | | 402.53 ±21.1 |
| Flowers | 115.04 ±2.12 | | 57.66 ±1.33 |
| LSUN | 104.91 ±14.3 | | 119.07 ±11.1 |
| SVHN | 15.04 ±0.48 | | 13.14 ±0.50 |
| ImageNet | 1375.88 ±30.5 | | 230.16 ±11.3 |
| accidents | 10.38 ±0.67 | | 19.03 ±1.79 |
| ad | 3.97 ±0.22 | | 11.75 ±0.84 |
| baudio | 11.47 ±0.18 | | 20.35 ±2.95 |
| bbc | 71.49 ±0.55 | | 80.06 ±3.29 |
| bnetflix | 17.61 ±0.24 | | 21.59 ±3.31 |
| book | 7.47 ±0.06 | | 13.09 ±3.13 |
| c20ng | 38.61 ±0.03 | | 64.73 ±10.3 |
| cr52 | 22.66 ±0.21 | | 31.68 ±2.27 |
| cwebkb | 43.63 ±0.36 | | 57.91 ±3.21 |
| dna | 32.34 ±0.07 | | 40.59 ±3.39 |
| jester | 16.38 ±0.23 | | 25.35 ±2.74 |
| kdd | 0.37 ±0.00 | | 0.43 ±0.04 |
| kosarek | 2.36 ±0.03 | | 2.96 ±0.15 |
| moviereview | 102.20 ±0.80 | | 131.58 ±13.0 |
| msnbc | 1.58 ±0.09 | | 1.96 ±0.02 |
| nltcs | 1.05 ±0.00 | | 1.32 ±0.02 |
| plants | 2.65 ±0.23 | | 3.55 ±0.37 |
| pumsb_star | 8.14 ±0.70 | | 20.08 ±5.77 |
| tmovie | 13.15 ±0.03 | | 19.06 ±1.22 |
| tretail | 2.20 ±0.00 | | 2.54 ±0.06 |
| voting | 51.00 ±0.97 | | 57.25 ±2.31 |
| | Downstream Accuracy (↑) | | |
| MNIST | 96.12 ±0.12 | | 90.39 ±0.07 |
| F-MNIST | 83.02 ±0.16 | | 81.04 ±0.21 |
| SVHN | 56.92 ±3.78 | | 23.98 ±0.35 |
| CIFAR | 47.14 ±0.37 | | 38.62 ±0.42 |
| Flowers | 9.87 ±0.65 | | 12.84 ±0.23 |
| ImageNet | 6.60 ±0.11 | | 2.97 ±0.14 |

**MCAR Corruptions**: APCs surpass their teacher in robustness against corruptions.

| | Teacher | → | Student |
|---|---|---|---|
| | MSE (↓) | | |
| MNIST | 35.38 ±1.74 | | 6.15 ±0.11 |
| F-MNIST | 39.80 ±1.10 | | 5.33 ±0.05 |
| CIFAR | 55.81 ±0.47 | | 19.61 ±0.10 |
| CelebA | 1095.90 ±14.0 | | 201.70 ±10.4 |
| Flowers | 107.75 ±11.8 | | 31.43 ±0.51 |
| LSUN | 254.75 ±2.39 | | 68.71 ±4.23 |
| SVHN | 41.01 ±0.15 | | 7.68 ±0.19 |
| ImageNet | 1286.51 ±10.4 | | 139.72 ±6.39 |
| accidents | 7.56 ±0.28 | | 7.76 ±0.37 |
| ad | 6.04 ±0.17 | | 5.68 ±0.23 |
| baudio | 7.44 ±0.01 | | 7.94 ±0.40 |
| bbc | 37.45 ±0.55 | | 39.04 ±1.20 |
| bnetflix | 13.49 ±0.77 | | 10.46 ±0.41 |
| book | 3.73 ±0.02 | | 6.12 ±0.85 |
| c20ng | 20.69 ±0.05 | | 29.86 ±1.85 |
| cr52 | 12.83 ±0.18 | | 15.98 ±0.61 |
| cwebkb | 22.94 ±0.15 | | 28.13 ±1.24 |
| dna | 16.46 ±0.17 | | 19.29 ±1.09 |
| jester | 17.44 ±0.10 | | 10.37 ±0.43 |
| kdd | 0.19 ±0.00 | | 0.22 ±0.01 |
| kosarek | 1.42 ±0.01 | | 1.58 ±0.05 |
| moviereview | 50.08 ±0.22 | | 62.55 ±3.66 |
| msnbc | 1.03 ±0.01 | | 1.07 ±0.04 |
| nltcs | 1.60 ±0.01 | | 1.12 ±0.01 |
| plants | 3.93 ±0.05 | | 2.55 ±0.05 |
| pumsb_star | 12.47 ±0.40 | | 9.22 ±2.80 |
| tmovie | 8.72 ±0.03 | | 8.81 ±0.33 |
| tretail | 1.23 ±0.00 | | 1.22 ±0.01 |
| voting | 122.24 ±52.8 | | 28.94 ±0.82 |
| | Downstream Accuracy (↑) | | |
| MNIST | 27.49 ±2.23 | | 87.77 ±0.12 |
| F-MNIST | 30.30 ±2.64 | | 79.41 ±0.24 |
| SVHN | 36.38 ±1.99 | | 22.93 ±0.29 |
| CIFAR | 33.53 ±0.50 | | 37.66 ±0.33 |
| Flowers | 4.53 ±0.35 | | 12.56 ±0.27 |
| ImageNet | 5.54 ±0.37 | | 2.97 ±0.15 |

# F   Embedding-based Out-of-Distribution Detection Using APCs

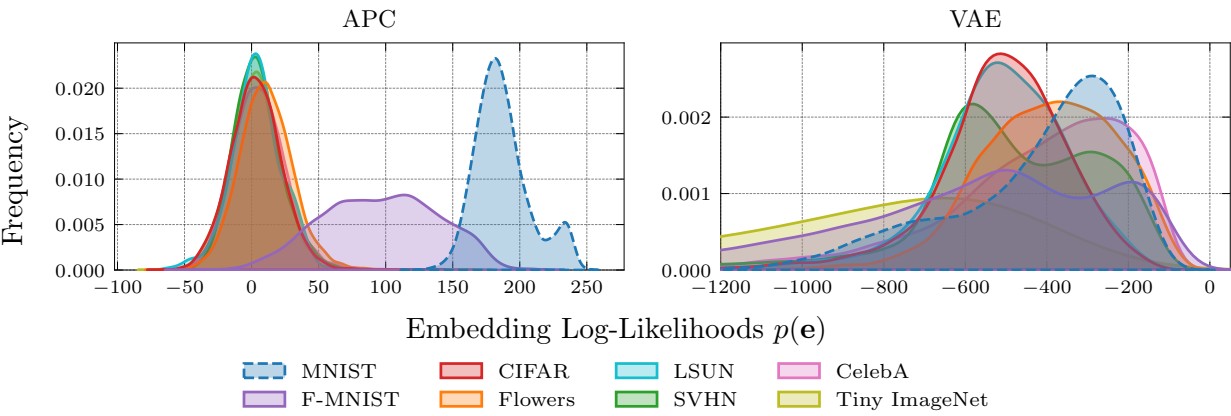

Figure 17: **While VAEs map in-distribution and out-of-distribution data to the same embedding range, APCs cleanly separate MNIST from other datasets.** As expected, we see some minor overlap with F-MNIST due to their similar pixel statistics, whereas all other datasets have close to no overlap with MNIST. APC and VAE models under the lens of out-of-distribution data: Models were trained on MNIST and embedding log-likelihoods $p(\mathbf{z})$ were evaluated for a range of out-of-distribution datasets. For APCs we can tractably obtain $p(\mathbf{z}) = \int p(\mathbf{x}, \mathbf{z}) \, d\mathbf{x}$ with marginalization. For VAEs, we need to compute an aggregate posterior from training dataset embeddings.

In addition to generative tasks, we can also investigate the empirical distribution of embedding likelihoods produced by the model. Such analysis could allow out-of-distribution (OOD) detection, enabling decoding processes or downstream applications to proactively *reject* embeddings that deviate significantly from the expected embedding distribution, as determined by a predefined threshold. As an initial exploration, Appendix Figure 17 compares the embedding log-likelihood distributions of APCs and VAEs for in-distribution (MNIST) and out-of-distribution datasets. We selected MNIST for our OOD experiments because its hand-drawn digit images represent a highly constrained and artificial distribution that differs from naturally occurring images. This distinctive characteristic, which MNIST shares with F-MNIST, establishes a clear distributional boundary that theoretically enables models to assign lower likelihood scores to out-of-distribution samples compared to in-distribution data points. As previously established, APCs allow us to evaluate the marginal embedding log-likelihood exactly and tractably. At the same time, VAEs require approximating this distribution using a post-hoc aggregate posterior from training data. As evident in Appendix Figure 17, APC embeddings separate in-distribution from out-of-distribution data in the likelihood space, with minimal overlap except for F-MNIST, which shares similar pixel statistics with MNIST. In contrast, VAE embeddings exhibit substantial overlap between in-distribution and out-of-distribution likelihood scores, making OOD detection almost impossible. This demonstrates that APCs' explicit probabilistic formulation provides strong reconstruction and representation learning capabilities and hints at out-of-distribution detection capabilities without requiring additional mechanisms or models. We leave further in-depth analysis on this topic to future work.

# G  Differentiable Sampling with SIMPLE: Details

---

**Algorithm 3** Sampling sum units with SIMPLE

---

**Require:** Normalized weights $\boldsymbol{\theta} \in [0,1]^D$ for sum unit $n$, where $D = |\text{in}(n)|$.
 1: **procedure** SIMPLE($\boldsymbol{\theta}$)
 2:    $\log \boldsymbol{\theta} \leftarrow \log(\boldsymbol{\theta})$                          ▷ Convert probabilities to log-probabilities
 3:    Draw $\mathbf{g} \sim \text{Gumbel}(0,1)^D$                    ▷ Draw independent Gumbel noise for each component
 4:    $\mathbf{z} \leftarrow \log \boldsymbol{\theta} + \mathbf{g}$                          ▷ Perturb log-probabilities with Gumbel noise
 5:    $d^* \leftarrow \arg\max_{j=1,\ldots,D} z_j$                ▷ Index of maximum perturbed probability (discrete sample)
 6:    $\mathbf{s} \leftarrow \text{one-hot}(d^*, D)$                       ▷ Construct a one-hot vector $\mathbf{s}$ where $s_{d^*} = 1$
 7:    **return** $(\mathbf{s} - \boldsymbol{\theta}).detach() + \boldsymbol{\theta}$        ▷ Return $\mathbf{s}$ for forward pass, but only pass gradients of $\boldsymbol{\theta}$ (*detach*)

---

We additionally improve the differentiable sampling approach for PCs originally proposed in Lang et al. (2022) by replacing the Gumbel-Softmax gradient estimator with SIMPLE (Ahmed et al., 2023) for $k$-subset sampling (with $k = 1$) when sampling from sum units as outlined in Appendix Algorithm 3. In contrast to Gumbel-Softmax, SIMPLE directly propagates the gradients through the unperturbed paramters $\boldsymbol{\theta}$. To quantitatively evaluate this modification, we conducted a controlled experiment comparing both gradient estimators on a synthetic task: learning a sum unit distribution with 32,64,128, and 256 inputs to match a known categorical ground truth distribution over 64 categories. We optimized sum unit weights by minimizing the MSE between one-hot encoded samples from both distributions, using AdamW with a learning rate of 0.01 for 1,000 iterations with batch size 64, repeating across 10 random seeds. As shown in Fig. 4, SIMPLE demonstrates faster convergence and higher accuracy in approximating the true distribution, as measured by KL. Notably, SIMPLE at iteration 100 achieves comparable performance to Gumbel-Softmax at iteration 1,000, and continues to improve beyond that point. Furthermore, SIMPLE demonstrates lower variance across different random initializations, indicating more consistent convergence behavior regardless of starting conditions. Upon convergence, SIMPLE achieves a KL of $0.0033 \pm 0.00065$, approximately $25\times$ lower than Gumbel-Softmax's $0.0817 \pm 0.00602$ for $D = 64$. These results validate our choice of SIMPLE as the gradient estimator for differentiable sampling in APCs, enabling more accurate and stable training.

# H    Analysis of Training and Inference Time

Table 9: Comparison of model computational time on the CelebA dataset, measured in milliseconds (ms) per batch on an NVIDIA A100 GPU. Inference time corresponds to a single reconstruction pass, and training time corresponds to a combined forward and backward pass. Both were measured with a batch size of 256.

| Model | Inference
Reconstruction (ms) | Training
Forward + Backward (ms) |
|---|---|---|
| APC | 64.9 | 236.2 |
| SPAE-ACT | 166.2 | 73.2 |
| SPAE-CAT | 165.8 | 73.2 |
| VAE | 24.3 | 43.5 |
| MIWAE | 1765.3 | 3933. |
| missForest | 1417.5 | – |

To provide a complete picture of the practical trade-offs between the evaluated models, we report their training (forward and backward pass) and inference (reconstruction) times for batch of size 256 on the CelebA dataset. The experiments were conducted on a single NVIDIA A100 GPU. We analyze the computational performance of APCs against the baselines. For MIWAE, we used 100 imputation iterations during training and inference to keep it feasible memory-wise, while the official implementation suggests 1000 or even up to 10000 iterations for the best performance.

**Inference Speed.**   VAEs leverage convolutional encoders, benefiting from highly optimized CUDA kernels that accelerate inference. In contrast, APCs and SPAEs employ circuit-based encoders whose operations are not as efficiently implemented on current hardware, leading to slower inference. SPAEs are particularly affected as both encoding and decoding are performed within the circuit. Iterative imputation methods, such as MIWAE and missForest, are substantially slower due to their multi-step, sequential procedures for estimating missing values.

**Training Speed.**   During training, APCs are slower than VAEs and SPAEs. This is attributed to the differentiable sampling mechanism, where the top-down pass currently presents a computational bottleneck. We note this is a limitation of the current implementation rather than a fundamental constraint. The top-down pass is characterized by extremely sparse tensors, only one input to each sum node is active (c.f. Appendix Algorithm 3 Line 6), which presents a clear opportunity for future optimization via a custom, sparsity-aware implementation. The missForest model does not have a training time per batch as it is a non-parametric method that builds its imputation model on-the-fly.

## I  Quantitative Evaluation of Sample Quality

To complement the qualitative analysis in Fig. 9, which demonstrates that APCs circumvent the visual artifacts common to samples from standard PCs, we provide a quantitative evaluation of sample quality. We assess generative fidelity using two standard metrics: the Fréchet Inception Distance (FID) (Heusel et al., 2017) and the Kernel Inception Distance (KID) (Sutherland et al., 2018). Scores are computed using the `clean-fid` (Parmar et al., 2022) Python package by comparing 50,000 generated samples against 50,000 samples from the respective test sets.

As reported in Appendix Table 10, APCs consistently achieve substantially lower (better) FID and KID scores across all datasets compared to a standard PC trained with maximum likelihood. This quantitative result corroborates our qualitative finding that integrating a neural decoder within the APC framework significantly enhances sample fidelity. While achieving state-of-the-art generative fidelity is beyond the scope of this work, we report these metrics primarily to quantify the improvement our hybrid architecture offers over traditional, purely circuit-based generation.

Table 10: **APCs achieve better FID and KID scores than vanilla PCs.** We report FID and KID for samples generated by APCs and a standard PC trained via maximum likelihood. Lower scores indicate better sample quality. The results confirm that the hybrid APC architecture significantly improves sample fidelity over purely circuit-based generation.

|  |  | MNIST | LSUN | CelebA | Flowers |
|---|---|---|---|---|---|
| FID | APC | $43.98_{\pm 1.06}$ | $268.35_{\pm 0.73}$ | $118.80_{\pm 0.89}$ | $170.49_{\pm 1.34}$ |
|  | PC | $137.65_{\pm 0.66}$ | $273.32_{\pm 0.23}$ | $353.02_{\pm 0.26}$ | $263.09_{\pm 0.60}$ |
| KID | APC | $0.0471_{\pm 0.0015}$ | $0.2712_{\pm 0.0012}$ | $0.1346_{\pm 0.0019}$ | $0.1725_{\pm 0.0022}$ |
|  | PC | $0.1493_{\pm 0.0010}$ | $0.3058_{\pm 0.0003}$ | $0.4607_{\pm 0.0003}$ | $0.2785_{\pm 0.0009}$ |

