# OpenReview forum: "Tractable Representation Learning with Probabilistic Circuits"
_TMLR — Accepted by TMLR_

### Review · Reviewer_fhbS · 2025-07-30

**Summary Of Contributions:**

The authors combine probabilistic circuits (PCs) with end-to-end autoencoding to learn data embeddings on the fly. This allows autoencoding PCs (APCs) to perform arbitrary marginalization and conditioning of latent variables from the data-conditional distribution, in addition to learning the joint distribution over embeddings and data. As a side effect, one can marginalize out missing variables in a relatively assumption-free manner. Overall the framework is a very handy extension of the PC model family, which can also be viewed as a (more flexible) extension to hierarchical VAEs. The empirical evaluation stands out as comprehensive and well thought out, with impressive results on corrupted data. I also want to commend the authors for the clarity of writing, notation, and presentation, which makes the paper a pleasure to read.

**Audience:**

Yes

**Audience Explanation:**

Probabilistic circuits (PCs) are a popular family of methods for tractable structured inference via deep learning. Algorithmic improvements and generalizations should be relevant to the community. See also summary of contributions.

**Claims And Evidence:**

Yes

**Claims Explanation:**

- The experiments feature different data types (tabular and images) and many different data sets.
- The experiments confirm APCs' robustness to data corruption and missingness.
- The experiments confirm APCs' ability to learn informative embeddings.
- The experiments indicate a clear improvement over baselines.
- The results are clear and easy to follow. The choice of baselines is well justified. The appendix contains sufficient detail. Code will be made public upon publication.

**Requested Changes:**

- The references regarding “tractability challenges in deep generative models” (p.2)  are outdated (the most recent one is from 2014) and should hint at many of the recent developments in tractable inference involving, for instance, normalizing flows [NFs; 1], VAE-NF combinations [2], and free-form flows [3]. These can also tractably compute densities, but, compared to PCs, lack the flexibility of PCs for joint, marginal, and conditional queries within the same model. Thus, the authors can try to clarify the benefits of PCs against this backdrop. Also, the flexibility of PCs can be a curse or a blessing, since there are infinite circuit designs that can be chosen with non-trivial consequences for performance. This limitation is only barely mentioned in the discussion and is probably deserving of more words.
- Missing value imputation can be flexibly tackled with the neural process (NP) model family [4, see also newer work on NPs], which is closely related to VAEs and should be discussed as well. In fact, a NP should have probably been included as a baseline in the image inpainting experiments (not required, but will definitely strengthen the paper).
- There are typos in the caption of Figure 3.
- VAEs can notoriously ignore the latent representation given sufficiently flexible decoders [see 5 for a convincing account], which is a direct consequence of the ELBO. Please, correct me if I am wrong, but I believe the same problem is inherent in the specification of the APC objective (Eq.1). Even if that's not the case, I think the topic deserves attention.
- The Kullback-Leibler divergence is conventionally abbreviated as KL throughout the literature. There is no need to invent new notation.

[1] Kobyzev, I., Prince, S. J., & Brubaker, M. A. (2020). Normalizing flows: An introduction and review of current methods. IEEE transactions on pattern analysis and machine intelligence, 43(11), 3964-3979.

[2] Horvat, C., & Pfister, J. P. (2021). Denoising normalizing flow. Advances in neural information processing systems, 34, 9099-9111.

[3] Draxler, F., Sorrenson, P., Zimmermann, L., Rousselot, A., & Köthe, U. (2024, April). Free-form flows: Make any architecture a normalizing flow. In International Conference on Artificial Intelligence and Statistics (pp. 2197-2205). PMLR.

[4] Garnelo, M., Schwarz, J., Rosenbaum, D., Viola, F., Rezende, D. J., Eslami, S. M., & Teh, Y. W. (2018). Neural processes. arXiv preprint arXiv:1807.01622.

[5] Zhao, S., Song, J., & Ermon, S. (2019). InfoVAE: Balancing learning and inference in variational autoencoders. In Proceedings of the aaai conference on artificial intelligence (Vol. 33, No. 01, pp. 5885-5892).

---

> ### Author Response · Authors · 2025-10-07
> **Rebuttal**
>
> We sincerely thank the reviewer for their detailed, insightful, and highly positive review. We are particularly encouraged by their praise for our framework as a "very handy extension of the PC model family," the "comprehensive and well thought out" empirical evaluation, and the overall "clarity of writing, notation, and presentation."
>
> We have carefully considered all suggestions and will revise the manuscript accordingly. Below, we address each point in detail.
>
> > The references regarding “tractability challenges in deep generative models” (p.2) are outdated ... and should hint at many of the recent developments in tractable inference involving, for instance, normalizing flows …
>
> Thank you for this excellent suggestion. We realize that the way we referenced at this position can be misunderstood. Our intention was to focus on circuit models that focus on tractability challenges (ACs, SPNs, PSDDs) and not list modern deep generative models at this point. Recent developments in tractable inference of deep generative models are highlighted in the Related Work section. Nevertheless, we agree that our discussion of tractable deep generative models will be strengthened by including the reviewer’s suggested recent advancements and have updated our manuscript.
>
> > Also, the flexibility of PCs can be a curse or a blessing, since there are infinite circuit designs ... This limitation is only barely mentioned...
>
> We also agree that the challenge of PC architecture design deserves more attention in general. We have expanded the paragraph on Encoder Structure in Section 3.2 with an additional discussion on the flexibility of PC structures and its implications.
>
> > Missing value imputation can be flexibly tackled with the neural process (NP) model family ... which is closely related to VAEs and should be discussed as well.
>
> We thank the reviewer for pointing out the relevant family of neural processes. We have added a discussion of NPs to our Related Work section (Section 4), acknowledging them as a powerful and principled framework for modeling distributions over functions.
>
> > VAEs can notoriously ignore the latent representation given sufficiently flexible decoders ... Please, correct me if I am wrong, but I believe the same problem is inherent in the specification of the APC objective (Eq.1). Even if that's not the case, I think the topic deserves attention.
>
> The reviewer is correct to question whether our framework is susceptible to the same posterior collapse issue that affects VAEs. We are happy to clarify why our objective function inherently mitigates this problem.
>
> The key difference lies in our inclusion of the joint log-likelihood term,  $\mathcal{L}\_{\text{NLL}} = -\frac{1}{B} \sum_{i=1}^{B} \log p(x_i, z_i)$. While the VAE's ELBO objective only encourages the model to learn a useful posterior via the reconstruction term, our $\mathcal{L}\_{\text{NLL}}$ term directly forces the PC encoder to model a high-likelihood joint distribution over data $x$ and embeddings $z$. If the embedding $z$ were to become uninformative or independent of $x$ (i.e., $p(z | x)$ collapses to $p(z)$), the joint probability $p(x, z)$ would be low, directly penalizing the model. Our ablation study in Table 5 empirically confirms the importance of this term: removing $\mathcal{L}\_{\text{NLL}}$ (the row "w/o $p(x, z)$") leads to a significant degradation in performance, underscoring its role in learning high-quality representations. We have added a paragraph to Section 3.4 to clarify this distinction from the standard VAE objective.
>
> > There are typos in the caption of Figure 3.
> > The Kullback-Leibler divergence is conventionally abbreviated as KL throughout the literature. There is no need to invent new notation.
>
> We thank the reviewer for catching the typos. We agree that "KL" is the standard abbreviation, and have replaced all instances of "KLD" with "KL" throughout the manuscript for consistency with community norms.
>
> —
>
> We hope these revisions and clarifications have fully addressed the reviewer's concerns. We are grateful for the constructive feedback, which has significantly improved our paper. We would be happy to provide further clarification if needed.

---

### Review · Reviewer_Cgbz · 2025-08-31

**Summary Of Contributions:**

### Summary of Contributions

The paper introduces Autoencoding Probabilistic Circuits (APCs), a framework for representation learning that explicitly integrates embeddings into probabilistic circuits (PCs). Unlike prior PC-based approaches such as SPAE or neural autoencoders such as VAEs, APCs

1. Model explicit probabilistic embeddings by extending PCs to jointly represent data and embedding variables.
2. Handle missing data natively through tractable inference and marginalization, without requiring imputation.
3. Enable hybrid training by coupling a PC encoder with a neural decoder, combining tractable probabilistic reasoning with the expressiveness of deep networks.
4. Introduce differentiable sampling (via SIMPLE) for end-to-end training of the encoder–decoder architecture.
5. Demonstrate empirical advantages over VAEs, MIWAE, SPAE, and imputation baselines across image and tabular datasets, particularly in robustness to missing data, embedding quality for downstream tasks, and data-free knowledge distillation.

### Key Strengths

* Novel formulation: APCs treat embeddings as first-class random variables within PCs, a principled and tractable approach absent in prior work.
* Robustness: Strong empirical performance under missing data (MCAR and MAR), where neural autoencoders degrade severely.
* Hybrid architecture: Combines the interpretability and tractability of PCs with the flexibility of neural decoders.
* Differentiable training: Incorporation of SIMPLE leads to stable and effective end-to-end optimization.
* Versatility: Supports both direct training and data-free knowledge distillation from pretrained models.
* Thorough evaluation: Results span reconstruction, downstream classification, embedding quality, ablations, and distillation across multiple datasets.

### Key Weaknesses

* Heuristic encoder structures: Placement of embedding variables in the PC encoder is somewhat ad hoc; more principled structure learning could improve performance.
* Scaling concerns: Current experiments focus on moderate-size datasets; unclear if APCs scale to the largest modern benchmarks such as ImageNet-scale.
* Comparisons limited: Main baselines are VAEs, MIWAE, SPAE, and imputation methods; stronger recent contenders such as diffusion-based autoencoders or masked autoencoders are not included.
* Complexity of implementation: Training APCs requires specialized PC infrastructure, potentially limiting accessibility and adoption.
* Generative fidelity gap: While robust, APC reconstructions may still lag behind state-of-the-art neural generative models on complete non-corrupted data.

**Additional Comments:**

N/A

**Audience:**

Yes

**Audience Explanation:**

The findings of this paper would be of interest to the TMLR audience because they advance representation learning with probabilistic circuits, a relatively underexplored yet important direction in tractable probabilistic modeling. The proposed APC framework combines the strengths of probabilistic circuits and neural networks, addressing key challenges such as robustness to missing data, explicit probabilistic embeddings, and data-free knowledge distillation. These contributions are relevant to researchers in machine learning who care about tractable inference, hybrid neural–probabilistic methods, and robust representation learning. Given that TMLR’s audience includes both theoretical and applied ML researchers, the novelty, empirical validation, and potential for follow-up work make this study appealing and timely.

**Broader Impact Concerns:**

The paper focuses on methodological advances in representation learning using probabilistic circuits and does not raise direct ethical risks such as those associated with generative text or image models that can be misused. However, there are some broader considerations worth noting. Since the method demonstrates strong robustness to missing data, one potential application area is in sensitive domains such as healthcare, finance, or social science datasets where incomplete records are common. In these settings, model misuse or overreliance on automatically inferred embeddings could lead to downstream decisions that impact individuals. It would therefore be important to include a broader impact statement acknowledging both the promise of APCs in improving robustness and the risks of deploying such models without careful domain-specific validation, fairness auditing, and transparency. Additionally, the requirement for specialized probabilistic circuit infrastructure might limit accessibility to only well-resourced research groups, raising equity concerns in who can practically adopt and extend this work.

**Claims And Evidence:**

Yes

**Claims Explanation:**

The submission provides extensive empirical evidence that directly supports its main claims. The authors evaluate APCs on a wide range of datasets (both image and tabular) and compare against multiple relevant baselines, including VAEs, MIWAE, SPAE, and imputation methods. They demonstrate consistent improvements in reconstruction quality, robustness under missing data, and downstream classification accuracy. The paper also includes qualitative results such as reconstructions and embedding visualizations, along with ablation studies that isolate the contribution of each design choice. These experiments collectively confirm the advantages claimed for APCs. While broader comparisons to more recent models would further strengthen the case, the evidence presented is clear, convincing, and sufficient to substantiate the core contributions.

**Requested Changes:**

### Critical Changes (needed for stronger acceptance)

1. Expand baseline comparisons: include evaluations against more recent and competitive representation learning methods such as masked autoencoders (MAEs) and diffusion-based autoencoders. This would better situate APCs in the current landscape and strengthen the empirical claims.
2. Clarify encoder design choices: the placement of embedding variables within the probabilistic circuit encoder is currently heuristic. More justification or principled reasoning for the chosen structures would make the approach clearer and more convincing.

### Recommended (non-critical, but would strengthen the work)

1. Scaling experiments: provide preliminary results or discussion on scaling APCs to larger and more complex datasets (for example ImageNet-scale). This would address concerns about applicability to modern large-scale benchmarks.

2. Analysis of computational cost: compare training and inference efficiency of APCs with neural autoencoders and other baselines, to give readers a clearer understanding of the trade-offs involved.

3. Discussion of limitations: expand on potential weaknesses of APCs, such as reliance on specialized PC infrastructure and challenges of broader adoption, to present a more balanced perspective.

4. Qualitative insights: while reconstructions and t-SNE plots are helpful, adding case studies or examples from real-world tabular domains (for example medical data with missing values) would highlight practical benefits.

---

> ### Author Response · Authors · 2025-10-07
> **Rebuttal**
>
> We sincerely thank the reviewer for their thoughtful and constructive feedback. We are particularly grateful for the positive assessment of our work, especially for recognizing the key strengths of our proposed Autoencoding Probabilistic Circuits (APCs), including the novel formulation of explicit probabilistic embeddings, the strong empirical robustness to missing data, the versatile hybrid architecture, and the thoroughness of our evaluation.
>
> We will now address the specific weaknesses and requested changes point by point. We believe that incorporating these suggestions will further strengthen the paper.
>
> **Baseline comparisons**
>
> We thank the reviewer for this important suggestion.
>
> > Main baselines are VAEs, MIWAE, SPAE, and imputation methods; stronger recent contenders such as diffusion-based autoencoders or masked autoencoders are not included.
>
> We agree that positioning APCs next to modern advancements is important. Our choice of baselines (VAE, MIWAE, SPAE) was motivated by their direct conceptual alignment with our work: they are probabilistic autoencoders that learn a global latent representation of the input. However, we acknowledge that discussing the relationship with other paradigms like MAEs and diffusion models will clarify the specific contributions of APCs.
>
> - Masked Autoencoders: As already discussed in our "Related Work" section, MAEs are designed for self-supervised pre-training of Transformers by learning to reconstruct randomly masked input patches from the visible ones. Their core mechanism is distinct from ours in two key ways: (1) MAEs do not produce a single, holistic embedding for the entire input instance. Instead, they operate on patch-level representations. This makes them unsuitable for tasks requiring a global, low-dimensional code, which is the primary goal of traditional autoencoders and our APC framework. (2) Masking in MAEs is a pre-defined, structured part of the training objective, not a test of robustness to arbitrary, unforeseen missing data patterns at inference time, which is the setting we are investigating.
>
> - Diffusion-based Autoencoders: These models excel at high-fidelity generation and reconstruction on complete data. However, their primary focus is on generative quality, often at the cost of tractable density evaluation and principled probabilistic inference. As we show in our NVAE experiments (Section 5.4), which we chose as a representative for modern advancements in VAEs, even state-of-the-art, scaled-up VAEs, fail on the core task of handling missing data. The primary contribution of APCs is not to outperform SOTA generative models on reconstruction fidelity with complete data, but rather to provide an autoencoding framework with tractable probabilistic inference and robustness, which remains a significant challenge for these fully neural-based autoencoder. As we outline in our conclusion section, future work may focus on adapting modern advancements from VAEs to APCs.
>
>
> **Heuristic encoder structures**
>
> > Placement of embedding variables in the PC encoder is somewhat ad hoc; more principled structure learning could improve performance.
>
> We agree that the PC encoder structure is a critical design choice. As discussed in our conclusion section, future work will focus on more strategic placements of embeddings RVS, e.g. distributing them across different circuit levels, leading to low-level and high-level representations and hierarchical structures in the embeddings space.
>
>
> **Complexity of implementation**
>
> > Training APCs requires specialized PC infrastructure, potentially limiting accessibility and adoption.
>
> We want to clarify that APCs, and in particular their PC encoder components, are implemented like standard neural network modules using off-the-shelf libraries such as PyTorch (used in our experiments), JAX, or TensorFlow. This ensures that APCs can run on all common deep learning hardware accelerators (GPUs, TPUs, IPUs) and do not require specialized infrastructure.
>
>
> **Scaling experiments**
>
> > Provide preliminary results or discussion on scaling APCs to larger and more complex datasets (for example ImageNet-scale). This would address concerns about applicability to modern large-scale benchmarks.
>
> While we agree with the reviewer, scaling PCs to arbitrary large architectures is an orthogonal research problem that we did not aim to tackle in this work.

---

> ### Author Response · Authors · 2025-10-07
> **Rebuttal (continued)**
>
> **Analysis of computational cost**
>
> > compare training and inference efficiency of APCs with neural autoencoders and other baselines, to give readers a clearer understanding of the trade-offs involved.
>
> We agree with the reviewer, that a direct benchmark of a forward pass will give the reader additional insights when comparing APCs with other models. Below we provide a table with forward pass time measurements on CelebA with a batch-size of 256, which will we have included in the new revision of the manuscript (Appendix H):
>
> | Model      | Inference (Reconstruction, ms) | Training (Forward + Backward, ms) |
> |:-----------|-------------------------------:|----------------------------------:|
> | APCs  |                           64.9 |                             236.2 |
> | SPAE-ACT   |                          166.2 |                              73.2 |
> | SPAE-CAT   |                          165.8 |                              73.2 |
> | VAE        |                           24.3 |                              43.5 |
> | MIWAE      |                         1765.3 |                            3933.0 |
> | missForest |                         1417.5 |                                -- |
>
> ---
>
> We thank the reviewer for their constructive feedback, which has strengthened the manuscript.

---

### Review · Reviewer_jKFS · 2025-10-10

**Summary Of Contributions:**

This paper proposes a representation learning framework, autoencoding probabilistic circuits, that models a joint distribution p_C(X, Z) over data X and embeddings Z with a probabilistic circuit. Embeddings are obtained by tractable conditional inference z ~ p_C(Z | X=x) which naturally supports arbitrary missingness at test time without imputation. They use a neural net as the decoder.

At APC sum nodes they replace the Gumbel-Softmax with SIMPLE from Ahmed et al. 2023 to improve gradient quality. They introduce a composite loss with 1) reconstruction error via the neural decoder, 2) a cheap KL regularizer on the embeddings that leverages their independence in the sampling-induced tree and 3) a joint data-embedding log-likelihood term log p_C(x,z) to mitigate posterior collapse. APCs are evaluated on 8 image datasets and 20 binary tabular datasets from the Density Estimation Benchmark Datasets (https://github.com/UCLA-StarAI/Density-Estimation-Datasets). They show that APCs obtain lower MSE / higher SSIM reconstructions under MCAR and MAR corruptions than SPAE, VAE, MIWAE, and missForest. They also show that APC embeddings remain linearly separable (as measured by logistic regression probes) under far more noise than other methods. They conduct ablations on MNIST and CIFAR for the key APC components and show that each makes an important contribution to reconstruction quality and linear probe accuracy. They also show that APCs can distill from pretrained VAEs and improve the robustness of reconstruction and linear probe accuracy under data corruptions.

**Additional Comments:**

Great work!

**Audience:**

Yes

**Audience Explanation:**

Researchers in tractable probabilistic modeling / probabilistic circuits, representation learning and hybrid probabilistic-neural architectures will find APCs directly relevant. The work is also relevant to those studying learning with missing data.

On the applied side, the ability to natively handle missingness and perform exact probabilistic queries makes this interesting for healthcare/EHR, survey/financial tabular modeling, remote sensing, and multimodal settings where partial observations are common.

**Claims And Evidence:**

Yes

**Claims Explanation:**

Claim 1. APCs enable tractable, exact probabilistic embeddings and native handling of missing inputs.
- Evidence: an explicit procedure to do so in algorithm 1 and a discussion of marginalizing X_m instead of imputing in Section 3.1.
- Verdict: Sufficient, especially since the mechanism is standard for PCs.

Claim 2: SIMPLE improves gradient estimation for differentiable PC sampling over Gumbel-Softmax.
- Evidence: synthetic task with KL to a ground-truth categorical distribution. Figure 4 and Appendix G show faster convergence, lower variance, lower final KL.
- Verdict: partially sufficient. The synthetic result is compelling but there is no comparison for APC training on real datasets. A SIMPLE vs. Gumbel-Softmax ablation in Table 1 would strengthen this claim.

Claim 3: APCs achieve lower recon error than SPAE, VAE, MIWAE and missForest baselines under MCAR and MAR missingness.
- Evidence: MCAR curves (Fig 5) and SSIM curves (Fig 12). Aggregate results under various MCAR levels in Tables 2, 3, 5 and MAR styles in Table 4. Visual comparisons of model samples in Figures 6 and 13-16.
- Verdict: Sufficient. Strong breadth (8 image + 20 tabular datasets), same neural decoder across APC/VAE/MIWAE makes for fair comparisons.

Claim 4: APC embeddings are more robust for downstream tasks under missingness than neural AEs
- Evidence: logistic regression accuracy curves across datasets in Figure 7. t-SNE projections in Figure 8.
- Verdict: sufficient. Consistent trends across datasets.

Claim 5: Each component of APCs (differentiable sampling, neural decoder, KL on Z, joint log p(x,z)) contributes meaningfully to performance.
- Evidence: Table 1 ablates the four components on MNIST/CIFAR under full-evidence and MCAR missingness.
- Verdict: Sufficient. Clear ablation results.

Claim 6: Neural decoding beats circuit-only decoding.
- Evidence: APC_pc vs APC with neural decoder in Table 6 (images) and Table 7 (tabular).
- Verdict: sufficient. The difference is large and consistent.

Claim 7: APCs effectively distill from a VAE without access to training data and are more robust to missingness than the teacher.
- Evidence: Figure 11 and Table 8 show that an APC student performs comparably to the VAE teacher on full evidence, but substantially better under missingness. t-SNE plots in Fig 11 under high missingness add qualitative evidence.
- Verdict: sufficient. Nicely demonstrated on multiple datasets with quantitative and qualitative evidence.

Claim 8: Modern high-capacity VAEs fail under heavy missingness.
- Evidence: Figure 10 shows failure modes for zero-imputation with VAEs and limited gains from MCAR finetuning.
- Verdict: insufficient. The evidence only covers one model family and one training recipe with zero-imputation. The claim should be softened or supplemented with masked-modeling or missingness-aware baselines.

Claim 9: APCs avoid typical PC sampling artifacts for images.
- Evidence: Qualitative samples in Figure 9 compare APC vs vanilla PC
- Verdict: qualitatively sufficient. A FID-like metric (where applicable) would strengthen the evidence.

Preliminary claim:

Claim 10: OOD detection can be done using APC embedding likelihoods.
- Evidence: Figure 17 (MNIST in-distribution vs several OODs) shows separation for APCs vs overlap for VAEs.
- Verdict: indeed preliminary (but interesting!). For this to be a central claim it needs quantitative metrics (e.g. AUROC) and more diverse datasets.

**Requested Changes:**

1. A SIMPLE vs. Gumbel-Softmax ablation in Table 1. Consider measuring gradient variance.
2. Claim 8 (above) should be softened or supplemented with masked-modeling or missingness-aware baselines.
3. Add a FID metric where applicable to Fig 9.

---

> ### Author Response · Authors · 2025-10-15
> **Rebuttal**
>
> We sincerely thank the reviewer for their exceptionally thorough and insightful feedback. We are grateful for the structured breakdown of our claims and the clear, actionable suggestions for improvement. The reviewer's positive assessment is very encouraging, and their requested changes have helped us to strengthen the paper.
>
> We have incorporated the suggested revisions. Below, we detail the changes corresponding to each point:
>
> 1. **Ablation Study on SIMPLE vs. Gumbel-Softmax:**
>     As requested, we have expanded our main ablation study in Table 1 to include a direct comparison between SIMPLE and Gumbel-Softmax within the full APC framework.
> | Dataset | Method | MSE (Full Evi.) ↓ | MSE (MCAR) ↓ | DS-Acc (Full Evi.) ↑ | DS-Acc (MCAR) ↑ |
> |:---|:---|---:|---:|---:|---:|
> | **MNIST** | Gumbel-Softmax | 7.60 ± 0.17 | 10.07 ± 0.21 | 86.62 ± 0.29 | 81.97 ± 0.63 |
> | | SIMPLE | 4.13 ± 0.10 | 5.06 ± 0.05 | 88.32 ± 0.14 | 86.85 ± 0.11 |
> | **CIFAR** | Gumbel-Softmax | 17.04 ± 0.19 | 28.23 ± 0.24 | 35.47 ± 0.23 | 34.13 ± 0.09 |
> | | SIMPLE | 16.29 ± 0.14 | 20.64 ± 0.08 | 37.61 ± 0.27 | 36.90 ± 0.14 |
>
>
> 2. **Softening the Claim Regarding Modern VAEs:**
>     We thank the reviewer for their point of view on this topic. We acknowledge that our original phrasing in Section 5.4 can be interpreted as too strong. Our intent was to investigate whether model scale alone is a sufficient condition for robustness, rather than to make a generalizing claim about all modern VAEs. We have revised the wording in Section 5.4 to soften our claim, explicitly framing the experiment as a preliminary study on a representative high-capacity model (NVAE).
>
> 3. **Quantitative Evaluation of Sample Quality (FID/KID):**
>     We have added results for both the FID and KID scores for APCs and PC on the datasets in Figure 9 in Appendix I, now also quantitatively confirming the visual improvements of APCs over PCs.
>
>     | | | **MNIST** | **LSUN** | **CelebA** | **Flowers** |
>     |:---|:---|:---:|:---:|:---:|:---:|
>     | **FID** | APC | 43.98 ± 1.06 | 268.35 ± 0.73 | 118.80 ± 0.89 | 170.49 ± 1.34 |
>     | | PC | 137.65 ± 0.66 | 273.32 ± 0.23 | 353.02 ± 0.26 | 263.09 ± 0.60 |
>     | **KID** | APC | 0.0471 ± 0.0015 | 0.2712 ± 0.0012 | 0.1346 ± 0.0019 | 0.1725 ± 0.0022 |
>     | | PC | 0.1493 ± 0.0010 | 0.3058 ± 0.0003 | 0.4607 ± 0.0003 | 0.2785 ± 0.0009 |
>
>
> Once again, we thank the reviewer for their constructive feedback, which has improved the rigor and clarity of our manuscript. We have uploaded the revised manuscript.

---

> > ### Comment · Reviewer_jKFS · 2025-10-23
> > **Great rebuttal**
> >
> > Thanks for adding the ablation and FID/KID metrics! I think the ablation makes the impact of SIMPLE vs Gumbel-Softmax more clear and quantitative. Similar sentiment for the FID/KID measurements. Clearly this additional evidence aligns with your original claims - nice work! I vote for accepting this paper.

---

### Decision · Action_Editor_G22S · 2025-11-18

**Recommendation:** Accept as is

**Audience:**

Yes

**Audience Explanation:**

Yes. The paper is of clearly interest to TMLR readers working on probabilistic circuits, representation learning, and handling missing data—active areas within the community. APCs offer a principled way to integrate probabilistic embeddings and tractable inference into autoencoding architectures, which is relevant to both theoretical and applied ML researchers. The results and methodology align well with topics regularly discussed in TMLR.

**Claims And Evidence:**

Yes

**Claims Explanation:**

Yes. The paper’s main claims are clearly supported by the evidence. The authors give a precise description of APCs and show, through well-designed experiments, that each component of the method works as intended. Their claims about robustness to missing data, improved embeddings, and the benefits are backed by quantitative results across many datasets and by targeted ablation studies. The additional experiments added after the rebuttal (SIMPLE vs. Gumbel-Softmax, FID/KID) further strengthen the evidence. Overall, the empirical results consistently validate the core contributions.